# Learning Rationalizable Equilibria in Multiplayer Games

**Yuanhao Wang**[*1], **Dingwen Kong**[*2], **Yu Bai**[3], **Chi Jin**[1]
[1]Princeton University, [2]Peking University, [3]Salesforce Research
`yuanhao@princeton.edu, dingwenk@pku.edu.cn`
`yu.bai@salesforce.com, chij@princeton.edu`

## Abstract

A natural goal in multi-agent learning is to learn *rationalizable* behavior, where players learn to avoid any Iteratively Dominated Action (IDA). However, standard no-regret based equilibria-finding algorithms could take exponential samples to find such rationalizable strategies. In this paper, we first propose a simple yet sample-efficient algorithm for finding a rationalizable action profile in multi-player general-sum games under bandit feedback, which substantially improves over the results of Wu et al. (2021). We further develop algorithms with the first efficient guarantees for learning rationalizable Coarse Correlated Equilibria (CCE) and Correlated Equilibria (CE). Our algorithms incorporate several novel techniques to guarantee the elimination of IDA and no (swap-)regret simultaneously, including a correlated exploration scheme and adaptive learning rates, which may be of independent interest. We complement our results with a sample complexity lower bound showing the sharpness of our guarantees.

## 1 Introduction

A common objective in multi-agent learning is to find various *equilibria*, such as Nash equilibria (NE), correlated equilibria (CE) and coarse correlated equilibria (CCE). Generally speaking, a player in equilibrium lacks incentive to deviate assuming conformity of other players to the same equilibrium. Equilibrium learning has been extensively studied in the literature of game theory and online learning, and no-regret based learners can provably learn approximate CE and CCE with both computational and statistical efficiency (Stoltz, 2005; Cesa-Bianchi & Lugosi, 2006).

However, not all equilibria are created equal. As shown by Viossat & Zapechelnyuk (2013), a CCE can be entirely supported on dominated actions—actions that are worse off than some other strategy in all circumstances—which rational agents should apparently never play. Approximate CE also suffers from a similar problem. As shown by Wu et al. (2021, Theorem 1), there are examples where an $\epsilon$-CE always plays iteratively dominated actions—actions that would be eliminated when iteratively deleting strictly dominated actions—unless $\epsilon$ is exponentially small. It is also shown that standard no-regret algorithms are indeed prone to finding such seemingly undesirable solutions (Wu et al., 2021). The intrinsic reason behind this is that CCE and approximate CE may not be *rationalizable*, and existing algorithms can indeed fail to find rationalizable solutions.

Different from equilibria notions, rationalizability (Bernheim, 1984; Pearce, 1984) looks at the game from the perspective of a single player without knowledge of the actual strategies of other players, and only assumes common knowledge of their rationality. A rationalizable strategy will avoid strictly dominated actions, and assuming other players have also eliminated their dominated actions, iteratively avoid strictly dominated actions in the subgame. Rationalizability is a central solution concept in game theory (Osborne & Rubinstein, 1994) and has found applications in auctions (Battigalli & Siniscalchi, 2003) and mechanism design (Bergemann et al., 2011).

If an (approximate) equilibrium only employs rationalizable actions, it would prevent irrational behavior such as playing dominated actions. Such equilibria are arguably more reasonable than

---
[*]Equal contribution.

unrationalizable ones, and constitute a stronger solution concept. This motivates us to consider the following open question:

*Can we efficiently learn equilibria that are also rationalizable?*

Despite its fundamental role in multi-agent reasoning, rationalizability is rarely studied from a learning perspective until recently, with Wu et al. (2021) giving the first algorithm for learning rationalizable strategies from bandit feedback. However, the problem of learning rationalizable CE and CCE remains a challenging open problem. Due to the existence of unrationalizable equilibria, running standard CE or CCE learners will not guarantee rationalizable solutions. On the other hand, one cannot hope to first identify all rationalizable actions and then find an equilibrium on the subgame, since even determining whether an action is rationalizable requires exponentially many samples (see Proposition 2). Therefore, achieving rationalizability and approximate equilibria simultaneously is nontrivial and presents new algorithmic challenges.

In this work, we address the challenges above and give a positive answer to our main question. Our contributions can be summarized as follows:

- As a first step, we provide a simple yet sample-efficient algorithm for identifying a $\Delta$-rationalizable [1] action profile under bandit feedback, using only $\widetilde{O}\left(\frac{LNA}{\Delta^2}\right)$[2] samples in normal-form games with $N$ players, $A$ actions per player and a minimum elimination length of $L$. This greatly improves the result of Wu et al. (2021) and is tight up to logarithmic factors when $L = O(1)$.

- Using the above algorithm as a subroutine, we develop exponential weights based algorithms that can provably find $\Delta$-rationalizable $\epsilon$-CCE using $\widetilde{O}\left(\frac{LNA}{\Delta^2} + \frac{NA}{\epsilon^2}\right)$ samples, and $\Delta$-rationalizable $\epsilon$-CE using $\widetilde{O}\left(\frac{LNA}{\Delta^2} + \frac{NA^2}{\min\{\epsilon^2, \Delta^2\}}\right)$ samples. To the best of our knowledge, these are the first guarantees for learning rationalizable approximate CCE and CE.

- We also provide reduction schemes that find $\Delta$-rationalizable $\epsilon$-CCE/CE using black-box algorithms for $\epsilon$-CCE/CE. Despite having slightly worse rates, these algorithms can directly leverage the progress in equilibria finding, which may be of independent interest.

## 1.1 RELATED WORK

**Rationalizability and iterative dominance elimination.** Rationalizability (Bernheim, 1984; Pearce, 1984) is a notion that captures rational reasoning in games and relaxes Nash Equilibrium. Rationalizability is closely related to the iterative elimination of dominated actions, which has been a focus of game theory research since the 1950s (Luce & Raiffa, 1957). It can be shown that an action is rationalizable if and only if it survives iterative elimination of strictly dominated actions[3] (Pearce, 1984). There is also experimental evidence supporting iterative elimination of dominated strategies as a model of human reasoning (Camerer, 2011).

**Equilibria learning in games.** There is a rich literature on applying online learning algorithms to learning equilibria in games. It is well-known that if all agents have no-regret, the resulting empirical average would be an $\epsilon$-CCE (Young, 2004), while if all agents have no swap-regret, the resulting empirical average would be an $\epsilon$-CE (Hart & Mas-Colell, 2000; Cesa-Bianchi & Lugosi, 2006). Later work continuing this line of research include those with faster convergence rates (Syrgkanis et al., 2015; Chen & Peng, 2020; Daskalakis et al., 2021), last-iterate convergence guarantees (Daskalakis & Panageas, 2018; Wei et al., 2020), and extension to extensive-form games (Celli et al., 2020; Bai et al., 2022b;a; Song et al., 2022) and Markov games (Song et al., 2021; Jin et al., 2021).

**Computational and learning aspect of rationalizability.** Despite its conceptual importance, rationalizability and iterative dominance elimination are not well studied from a computational or learning perspective. For iterative strict dominance elimination in two-player games, Knuth et al. (1988) provided a cubic-time algorithm and proved that the problem is P-complete. The weak dominance version of the problem is proven to be NP-complete by Conitzer & Sandholm (2005).

---

[1]An action is $\Delta$-rationalizable if it survives iterative elimination of $\Delta$-dominated actions; c.f. Definition 1.

[2]Throughout this paper, we use $\widetilde{O}$ to suppress logarithmic factors in $N$, $A$, $L$, $\frac{1}{\Delta}$, $\frac{1}{\delta}$, and $\frac{1}{\epsilon}$.

[3]For this equivalence to hold, we need to allow dominance by mixed strategies, and correlated beliefs when there are more than two players. These conditions are met in the setting of this work.

Hofbauer & Weibull (1996) showed that in a class of learning dynamics which includes replicator dynamics — the continuous-time variant of Follow-The-Regularzied-Leader (FTRL), all iteratively strictly dominated actions vanish over time, while Mertikopoulos & Moustakas (2010) proved similar results for stochastic replicator dynamics; however, neither work provides finite-time guarantees. Cohen et al. (2017) proved that Hedge eliminates dominated actions in finite time, but did not extend their results to the more challenging case of iteratively dominated actions.

The most related work in literature is the work on learning rationalizable actions by Wu et al. (2021), who proposed the Exp3-DH algorithm to find a strategy mostly supported on rationalizable actions with a polynomial rate. Our Algorithm 1 accomplishes the same task with a faster rate, while our Algorithms 2 & 3 deal with the more challenging problems of finding $\epsilon$-CE/CCE that are also rationalizable. Although Exp3-DH is based on a no-regret algorithm, it does not enjoy regret or weighted regret guarantees and thus does not provably find rationalizable equilibria.

## 2 PRELIMINARY

An $N$-player normal-form game involves $N$ players whose action space are denoted by $\mathcal{A} = \mathcal{A}_1 \times \cdots \times \mathcal{A}_N$, and is defined by utility functions $u_1, \cdots, u_N : \mathcal{A} \to [0, 1]$. Let $A = \max_{i \in [N]} |A_i|$ denote the maximum number of actions per player, $x_i$ denote a mixed strategy of the $i$-th player (*i.e.*, a distribution over $\mathcal{A}_i$) and $x_{-i}$ denote a (correlated) mixed strategy of the other players (*i.e.*, a distribution over $\prod_{j \neq i} \mathcal{A}_j$). We further denote $u_i(x_i, x_{-i}) := \mathbb{E}_{a_i \sim x_i, a_{-i} \sim x_{-i}} u_i(a_i, a_{-i})$. We use $\Delta(S)$ to denote a distribution over the set $S$.

**Learning from bandit feedback** We consider the bandit feedback setting where in each round, each player $i \in [N]$ chooses an action $a_i \in \mathcal{A}_i$, and then observes a random feedback $U_i \in [0, 1]$ such that $\mathbb{E}[U_i | a_1, a_2, \cdots, a_n] = u_i(a_1, a_2, \cdots, a_n)$.

### 2.1 RATIONALIZABILITY

An action $a \in \mathcal{A}_i$ is said to be rationalizable if it could be the best response to some (possibly correlated) belief of other players' strategies, assuming that they are also rational. In other words, the set of rationalizable actions is obtained by iteratively removing actions that could never be a best response. For finite normal-form games, this is in fact equivalent to the iterative elimination of strictly dominated actions[4] (Osborne & Rubinstein, 1994, Lemma 60.1).

**Definition 1** ($\Delta$-Rationalizability). [5] *Define*

$$E_1 := \bigcup_{i=1}^N \left\{ a \in \mathcal{A}_i : \exists x \in \Delta(\mathcal{A}_i), \forall a_{-i}, \quad u_i(a, a_{-i}) \leq u_i(x, a_{-i}) - \Delta \right\},$$

*which is the set of $\Delta$-dominated actions for all players. Further define*

$$E_l := \bigcup_{i=1}^N \left\{ a \in \mathcal{A}_i : \exists x \in \Delta(\mathcal{A}_i), \forall a_{-i} \text{ s.t. } a_{-i} \cap E_{l-1} = \emptyset, u_i(a, a_{-i}) \leq u_i(x, a_{-i}) - \Delta \right\},$$

*which is the set of actions that would be eliminated by the $l$-th round. Define $L = \inf\{l : E_{l+1} = E_l\}$ as the **minimum elimination length**, and $E_L$ as the set of $\Delta$-**iteratively dominated actions ($\Delta$-IDAs)**. Actions in $\cup_{i=1}^n \mathcal{A}_i \setminus E_L$ are said to be $\Delta$-**rationalizable**.*

Notice that $E_1 \subseteq \cdots \subseteq E_L = E_{L+1}$. Here $\Delta$ plays a similar role as the reward gap for best arm identification in stochastic multi-armed bandits. We will henceforth use $\Delta$-rationalizability and survival of $L$ rounds of iterative dominance elimination (IDE) interchangeably[6]. Since one cannot eliminate all the actions of a player, $|E_L| \geq N$, which further implies $L \leq N(A-1) < NA$.

### 2.2 EQUILIBRIA IN GAMES

We consider three common learning objectives, namely Nash Equilibrium (NE), Correlated Equilibrium (CE) and Coarse Correlated Equilibrium (CCE).

---

[4]See, *e.g.*, the Diamond-In-the-Rough (DIR) games in Wu et al. (2021, Definition 2) for a concrete example of iterative dominance elimination.

[5]Here we slightly abuse the notation and use $\Delta$ to refer to both the gap and the probability simplex.

[6]Alternatively one can also define $\Delta$-rationalizability by the iterative elimination of actions that are never $\Delta$-best response, which is mathematically equivalent to Definition 1 (see Appendix A.1).

**Definition 2** (Nash Equilibrium). *A strategy profile* $(x_1, \cdots, x_N)$ *is an* $\epsilon$-*Nash equilibrium if*

$$u_i(x_i, x_{-i}) \geq u_i(a, x_{-i}) - \epsilon, \forall a \in \mathcal{A}_i, \forall i \in [N].$$

**Definition 3** (Correlated Equilibrium). *A correlated strategy* $\Pi \in \Delta(\mathcal{A})$ *is an* $\epsilon$-*correlated equilibrium if* $\forall i \in [N], \forall \phi : \mathcal{A}_i \to \mathcal{A}_i,$

$$\sum_{a_i \in \mathcal{A}_i, a_{-i} \in \mathcal{A}_{-i}} \Pi(a_i, a_{-i}) u_i(a_i, a_{-i}) \geq \sum_{a_i \in \mathcal{A}_i, a_{-i} \in \mathcal{A}_{-i}} \Pi(a_i, a_{-i}) u_i(\phi(a_i), a_{-i}) - \epsilon.$$

**Definition 4** (Coarse Correlated Equilibrium). *A correlated strategy* $\Pi \in \Delta(\mathcal{A})$ *is an* $\epsilon$-*CCE if* $\forall i \in [N], \forall a' \in \mathcal{A}_i,$

$$\sum_{a_i \in \mathcal{A}_i, a_{-i} \in \mathcal{A}_{-i}} \Pi(a_i, a_{-i}) u_i(a_i, a_{-i}) \geq \sum_{a_i \in \mathcal{A}_i, a_{-i} \in \mathcal{A}_{-i}} \Pi(a_i, a_{-i}) u_i(a', a_{-i}) - \epsilon.$$

When $\epsilon = 0$, the above definitions give exact Nash equilibrium, correlated equilibrium, and coarse correlated equilibrium, respectively. It is well known that $\epsilon$-NE are $\epsilon$-CE, and $\epsilon$-CE are $\epsilon$-CCE. Furthermore, we call an $\epsilon$-CCE/CE that only plays $\Delta$-rationalizable actions a.s. a $\Delta$-*rationalizable* $\epsilon$-*CCE/CE*.

### 2.3 CONNECTION BETWEEN EQUILIBRIA AND RATIONALIZABILITY

It is known that all actions in the support of an exact CE are rationalizable (Osborne & Rubinstein, 1994, Lemma 56.2). However, one can easily construct an exact CCE that is supported on dominated (hence, unrationalizable) actions (see *e.g.* Viossat & Zapechelnyuk (2013, Fig. 3)). One might be tempted to suggest that running a CE solver immediately finds a CE (and hence CCE) that is also rationalizable. However, the connection between CE and rationalizability becomes quite different when it comes to approximate equilibria, which are inevitable in the presence of noise. As shown by Wu et al. (2021, Theorem 1), an $\epsilon$-CE can be entirely supported on iteratively dominated action, unless $\epsilon = O(2^{-A})$. In other words, rationalizability is not guaranteed by running an approximate CE solver unless with an extremely high accuracy. Therefore, finding $\epsilon$-CE and CCE that are simultaneously rationalizable remains a challenging open problem.

Since NE is a subset of CE, all actions in the support of an (exact) NE would also be rationalizable. Unlike approximate CE, for $\epsilon < \text{poly}(\Delta, 1/N, 1/A))$, one can show that any $\epsilon$-Nash equilibrium is still mostly supported on rationalizable actions.

**Proposition 1.** *If* $x^* = (x_1^*, \cdots, x_N^*)$ *is an* $\epsilon$-*Nash with* $\epsilon < \frac{\Delta^2}{24N^2A}$, $\forall i$, $\Pr_{a \sim x_i^*}[a \in E_L] \leq \frac{2L\epsilon}{\Delta}$.

Therefore, for two-player zero-sum games, it is possible to run an approximate NE solver and automatically find a rationalizable $\epsilon$-NE. However, this method will induce a rather slow rate[7], and we will provide a much more efficient algorithm for finding rationalizable $\epsilon$-NE in Section 4.

## 3 LEARNING RATIONALIZABLE ACTION PROFILES

In order to learn a rationalizable CE/CCE, one might suggest identifying the set of all rationalizable actions, and then learn CE or CCE on this subgame. Unfortunately, as shown by Proposition 2, even the simpler problem of *deciding* whether one single action is rationalizable is statistically hard.

**Proposition 2.** *For* $\Delta < 0.1$, *any algorithm that correctly decides whether an action is* $\Delta$-*rationalizable with* 0.9 *probability needs* $\Omega(A^{N-1}\Delta^{-2})$ *samples.*

This negative result motivates us to consider an easier task: can we at least find one rationalizable action profile sample-efficiently? Formally, we say a action profile $(a_1, \ldots, a_N)$ is rationalizable if for all $i \in [N]$, $a_i$ is a rationalizable action. This is arguably one of the most fundamental tasks regarding rationalizability. For *mixed-strategy dominance solvable* games (Alon et al., 2021), the unique rationalizable action profile will be the unique NE and also the unique CE of the game. Therefore this easier task *per se* is still of practical importance.

In this section we answer this question in the affirmative. We provide a sample-efficient algorithm which finds a rationalizable action profile using only $\widetilde{O}\left(\frac{LNA}{\Delta^2}\right)$ samples. This algorithm will also serve as an important subroutine for algorithms finding rationalizable CCE/CE in the later sections.

---

[7]For two-player zero-sum games, the marginals of any CCE is an NE so NE can be found efficiently. This is not true for general games, where finding NE is computationally hard and takes $\Omega(2^N)$ samples.

---

**Algorithm 1** Iterative Best Response

---

1: **Initialization:** choose $a_i^{(0)} \in \mathcal{A}_i$ arbitrarily for all $i \in [N]$
2: **for** $l = 1, \cdots, L$ **do**
3:    **for** $i \in [N]$ **do**
4:       For all $a \in \mathcal{A}_i$, play $(a, a_{-i}^{(l-1)})$ for $M$ times, compute player $i$'s average payoff $\hat{u}_i(a, a_{-i}^{(l-1)})$
5:       Set $a_i^{(l)} \leftarrow \arg\max_{a \in \mathcal{A}_i} \hat{u}_i(a, a_{-i}^{(l-1)})$      // Computing the empirical best response
6: **return** $(a_1^{(L)}, \cdots, a_N^{(L)})$

---

The intuition behind this algorithm is simple: if an action profile $a_{-i}$ can survive $l$ rounds of IDE, then its best response $a_i$ (i.e., $\arg\max_{a \in \mathcal{A}_i} u_i(a, a_{-i})$) can survive at least $l+1$ rounds of IDE, since the action $a_i$ can only be eliminated after some actions in $a_{-i}$ are eliminated. Concretely, we start from an arbitrary action profile $(a_1^{(0)}, \ldots, a_N^{(0)})$. In each round $l \in [L]$, we compute the (empirical) best response of $a_{-i}^{(l-1)}$ for each $i \in [N]$, and use those best responses to construct a new action profile $(a_1^{(l)}, \ldots, a_N^{(l)})$. By constructing iterative best responses, we will end up with an action profile that can survive $L$ rounds of IDE, which means surviving any number of rounds of IDE according to the definition of $L$. The full algorithm is presented in Algorithm 1, for which we have the following theoretical guarantee.

**Theorem 3.** *With* $M = \left\lceil \frac{16\ln(LNA/\delta)}{\Delta^2} \right\rceil$, *with probability* $1-\delta$, *Algorithm 1 returns an action profile that is* $\Delta$-*rationalizable using a total of* $\widetilde{O}\left(\frac{LNA}{\Delta^2}\right)$ *samples.*

Wu et al. (2021) provide the first polynomial sample complexity results for finding rationalizable action profiles. They prove that the Exp3-DH algorithm is able to find a distribution with $1 - \zeta$ fraction supported on $\Delta$-rationalizable actions using $\widetilde{O}\left(\frac{L^{1.5} N^3 A^{1.5}}{\zeta^3 \Delta^3}\right)$ samples under bandit feedback[8].

Compared to their result, our sample complexity bound $\widetilde{O}\left(\frac{LNA}{\Delta^2}\right)$ has more favorable dependence on all problem parameters, and our algorithm will output a distribution that is fully supported on rationalizable actions (thus has no dependence on $\zeta$).

We further complement Theorem 3 with a sample complexity lower bound showing that the linear dependency on $N$ and $A$ are optimal. This lower bound suggests that the $\widetilde{O}\left(\frac{LNA}{\Delta^2}\right)$ upper bound is tight up to logarithmic factors when $L = O(1)$, and we conjecture that this is true for general $L$.

**Theorem 4.** *Even for games with* $L \leq 2$, *any algorithm that returns a* $\Delta$-*rationalizable action profile with* $0.9$ *probability needs* $\Omega\left(\frac{NA}{\Delta^2}\right)$ *samples.*

**Conjecture 5.** *The minimax optimal sample complexity for finding a* $\Delta$-*rationalizable action profile is* $\Theta\left(\frac{LNA}{\Delta^2}\right)$ *for games with minimum elimination length* $L$.

## 4    LEARNING RATIONALIZABLE COARSE CORRELATED EQUILIBRIA (CCE)

In this section we introduce our algorithm for efficiently learning rationalizable CCEs. The high-level idea is to run no-regret Hedge-style algorithms for every player, while constraining the strategy inside the rationalizable region. Our algorithm is motivated by the fact that the probability of playing a dominated action will decay exponentially over time in the Hedge algorithm for adversarial bandit under full information feedback (Cohen et al., 2017). The full algorithm description is provided in Algorithm 2, and here we explain several key components in our algorithm design.

**Correlated Exploration Scheme.** In the bandit feedback setting, standard exponential weights algorithms such as EXP3.IX require *importance sampling* and *biased estimators* to derive a high-probability regret bound (Neu, 2015). However, such bias could cause a dominating strategy to lose its advantage. In our algorithm we adopt a correlated exploration scheme, which essentially simulates full information feedback by bandit feedback using $NA$ samples. Specifically, at every time step $t$,

---

[8]Wu et al. (2021)'s result allows trade-off between variables via different choice of algorithmic parameters. However, a $\zeta^{-1}\Delta^{-3}$ factor is unavoidable regardless of choice of parameters.

---

**Algorithm 2** Hedge for Rationalizable $\epsilon$-CCE

---

1: $(a_1^\star, \cdots, a_N^\star) \leftarrow$ Algorithm 1
2: For all $i \in [N]$, initialize $\theta_i^{(1)}(\cdot) \leftarrow \mathbb{1}[\cdot = a_i^\star]$
3: **for** $t = 1, \cdots, T$ **do**
4:     **for** $i = 1, \cdots, N$ **do**
5:        For all $a \in \mathcal{A}_i$, play $(a, \theta_{-i}^{(t)})$ for $M_t$ times, compute player $i$'s average payoff $u_i^{(t)}(a)$
6:        Set $\theta_i^{(t+1)}(\cdot) \propto \exp\left(\eta_t \sum_{\tau=1}^{t} u_i^{(\tau)}(\cdot)\right)$
7: For all $t \in [T]$ and $i \in [N]$, eliminate all actions in $\theta_i^{(t)}$ with probability smaller than $p$, then renormalize the vector to simplex as $\bar{\theta}_i^{(t)}$
8: **output:** $\left(\sum_{t=1}^{T} \otimes_{i=1}^{n} \bar{\theta}_i^{(t)}\right)/T$

---

the players take turn to enumerate their action set, while the other players fix their strategies according to Hedge. For $i \in [N]$ and $t \geq 2$, we denote $\theta_i^{(t)}$ the strategy computed using Hedge for player $i$ in round $t$. Joint strategy $(a, \theta_{-i}^{(t)})$ is played to estimate player $i$'s payoff $u_i^{(t)}(a)$. It is important to note that such correlated scheme *does not* require any communication between the players—the players can schedule the whole process before the game starts.

**Rationalizable Initialization and Variance Reduction.** We use Algorithm 1, which learns a rationalizable action profile, to give the strategy for the first round. By carefully preserving the disadvantage of any iteratively dominated action, we keep the iterates inside the rationalizable region throughout the whole learning process. To ensure this for every iterate with high probability, a minibatch is used to reduce the variance of the estimator.

**Clipping.** In the final step, we clip all actions with small probabilities, so that iteratively dominated actions do not appear in the output. The threshold is small enough to not affect the $\epsilon$-CCE guarantee.

### 4.1 THEORETICAL GUARANTEE

In Algorithm 2, we choose parameters in the following manner:

$$\eta_t = \max\left\{\sqrt{\frac{\ln A}{t}}, \frac{4\ln(1/p)}{\Delta t}\right\}, M_t = \left\lceil \frac{64\ln(ANT/\delta)}{\Delta^2 t} \right\rceil, \text{ and } p = \frac{\min\{\epsilon, \Delta\}}{8AN}. \tag{1}$$

Note that our learning rate can be bigger than the standard learning rate in FTRL algorithms when $t$ is small. The purpose is to guarantee the rationalizability of the iterates from the beginning of the learning process. As will be shown in the proof, this larger learning rate will not hurt the final rate. We now state the theoretical guarantee for Algorithm 2.

**Theorem 6.** *With parameters chosen as in Eq.(1), after $T = \widetilde{O}\left(\frac{1}{\epsilon^2} + \frac{1}{\epsilon\Delta}\right)$ rounds, with probability $1 - 3\delta$, the output strategy of Algorithm 2 is a $\Delta$-rationalizable $\epsilon$-CCE. The total sample complexity is*

$$\widetilde{O}\left(\frac{LNA}{\Delta^2} + \frac{NA}{\epsilon^2}\right).$$

**Remark 7.** *Due to our lower bound (Theorem 4), an $\widetilde{O}(\frac{NA}{\Delta^2})$ term is unavoidable since learning a rationalizable action profile is an easier task than learning rationalizable CCE. Based on our Conjecture 5, the additional $L$ dependency is also likely to be inevitable. On the other hand, learning an $\epsilon$-CCE alone only requires $\widetilde{O}(\frac{A}{\epsilon^2})$ samples, where as in our bound we have a larger $\widetilde{O}(\frac{NA}{\epsilon^2})$ term. The extra $N$ factor is a consequence of our correlated exploration scheme in which only one player explores at a time. Removing this $N$ factor might require more sophisticated exploration methods and utility estimators, which we leave as future work.*

**Remark 8.** *Evoking Algorithm 1 requires knowledge of $L$, which may not be available in practice. In that case, an estimate $L'$ may be used in its stead. If $L' \geq L$ (for instance when $L' = NA$), we can recover the current rationalizability guarantee, albeit with a larger sample complexity scaling with $L'$. If $L' < L$, we can still guarantee that the output policy avoids actions in $E_{L'}$, which are, informally speaking, actions that would be eliminated with $L'$ levels of reasoning.*

### 4.1.1 OVERVIEW OF THE ANALYSIS

We give an overview of our analysis of Algorithm 2 below. The full proof is deferred to Appendix C.

**Step 1: Ensure rationalizability.** We will first show that rationalizability is preserved at each iterate, *i.e.*, actions in $E_L$ will be played with low probability across all iterates. Formally,

**Lemma 9.** *With probability at least $1 - 2\delta$, for all $t \in [T]$ and all $i \in [N]$, $a_i \in \mathcal{A}_i \cap E_L$, we have $\theta_i^{(t)}(a_i) \leq p$.*

Here $p$ is defined in (1). Lemma 9 guarantees that, after the clipping in Line 7 of Algorithm 2, the output correlated strategy be $\Delta$-rationalizable.

We proceed to explain the main idea for proving Lemma 9. A key observation is that the set of rationalizable actions, $\cup_{i=1}^n \mathcal{A}_i \setminus E_L$, is closed under best response—for the $i$-th player, as long as the other players continue to play actions in $\cup_{j \neq i} \mathcal{A}_j \setminus E_L$, actions in $\mathcal{A}_i \cap E_L$ will suffer from excess losses each round in an exponential weights style algorithm. Concretely, for any $a_{-i} \in (\prod_{j \neq i} \mathcal{A}_j) \setminus E_L$ and any iteratively dominated action $a_i \in \mathcal{A}_i \cap E_L$, there always exists $x_i \in \Delta(\mathcal{A}_i)$ such that

$$u_i(x_i, a_{-i}) \geq u_i(a_i, a_{-i}) + \Delta.$$

With our choice of $p$ in Eq. (1), if other players choose their actions from $\cup_{j \neq i} \mathcal{A}_j \setminus E_L$ with probability $1 - pAN$, we can still guarantee an excess loss of $\Omega(\Delta)$. It follows that

$$\sum_{\tau=1}^t u_i^{(\tau)}(x_i) - \sum_{\tau=1}^t u_i^{(\tau)}(a_i) \geq \Omega(t\Delta) - \text{Sampling Noise}.$$

However, this excess loss can be obscured by the noise from bandit feedback when $t$ is small. Note that it is crucial that the statement of Lemma 9 holds for all $t$ due to the inductive nature of the proof. As a solution, we use a minibatch of size $M_t = \widetilde{O}\left(\lceil \frac{1}{\Delta^2 t} \rceil\right)$ in the $t$-th round to reduce the variance of the payoff estimator $u_i^{(t)}$. The noise term can now be upper-bounded with Azuma-Hoeffding by

$$\text{Sampling Noise} \leq \widetilde{O}\left(\sqrt{\sum_{\tau=1}^t \frac{1}{M_t}}\right) \leq O(t\Delta),$$

Combining this with our choice of the learning rate $\eta_t$ gives

$$\eta_t \left( \sum_{\tau=1}^t u_i^{(\tau)}(x_i) - \sum_{\tau=1}^t u_i^{(\tau)}(a_i) \right) \gg 1. \tag{2}$$

By the update rule of the Hedge algorithm, this implies that $\theta_i^{(t+1)}(a_i) \leq p$, which enables us to complete the proof of Lemma 9 via induction on $t$.

**Step 2: Combine with no-regret guarantees.** Next, we prove that the output strategy is an $\epsilon$-CCE. For a player $i \in [N]$, the regret is defined as $\text{Regret}_T^i = \max_{\theta \in \Delta(\mathcal{A}_i)} \sum_{t=1}^T \langle u_i^{(t)}, \theta - \theta_i^{(t)} \rangle$. We can obtain the following regret bound by standard analysis of FTRL with changing learning rates.

**Lemma 10.** *For all $i \in [N]$, $\text{Regret}_T^i \leq \widetilde{O}\left(\sqrt{T} + \frac{1}{\Delta}\right)$.*

Here the additive $1/\Delta$ term is the result of our larger $\widetilde{O}(\Delta^{-1} t^{-1})$ learning rate for small $t$. It follows from Lemma 10 that $T = \widetilde{O}\left(\frac{1}{\epsilon^2} + \frac{1}{\Delta\epsilon}\right)$ suffices to guarantee that the correlated strategy $\frac{1}{T}\left(\sum_{t=1}^T \otimes_{i=1}^n \theta_i^{(t)}\right)$ is an $(\epsilon/2)$-CCE. Since $pNA = O(\epsilon)$, the clipping step only minorly affects the CCE guarantee and the clipped strategy $\frac{1}{T}\left(\sum_{t=1}^T \otimes_{i=1}^n \bar{\theta}_i^{(t)}\right)$ is an $\epsilon$-CCE.

### 4.2 APPLICATION TO LEARNING RATIONALIZABLE NASH EQUILIBRIUM

Algorithm 2 can also be applied to two-player zero-sum games to learn a rationalizable $\epsilon$-NE efficiently. Note that in two-player zero-sum games, the marginal distribution of an $\epsilon$-CCE is guaranteed to be a $2\epsilon$-Nash (see, *e.g.*, Proposition 9 in Bai et al. (2020)). Hence direct application of Algorithm 2 to a zero-sum game gives the following sample complexity bound.

**Corollary 11.** *In a two-player zero-sum game, the sample complexity for finding a $\Delta$-rationalizable $\epsilon$-Nash with Algorithm 2 is $\widetilde{O}\left(\frac{LA}{\Delta^2} + \frac{A}{\epsilon^2}\right)$.*

This result improves over a direct application of Proposition 1, which gives $\widetilde{O}\left(\frac{A^3}{\Delta^4} + \frac{A}{\epsilon^2}\right)$ sample complexity and produces an $\epsilon$-Nash that could still take unrationalizable actions with positive probability.

---

**Algorithm 3** Adaptive Hedge for Rationalizable $\epsilon$-CE

---

1: $(a_1^\star, \cdots, a_N^\star) \leftarrow$ `Algorithm 1`
2: For all $i \in [N]$, initialize $\theta_i^{(1)} \leftarrow (1 - |\mathcal{A}_i|p)\mathbb{1}[\cdot = a_i^\star] + p\mathbb{1}$
3: **for** $t = 1, 2, \ldots, T$ **do**
4:   **for** $i = 1, 2, \ldots, N$ **do**
5:     For all $a \in \mathcal{A}_i$, play $(a, \theta_{-i}^{(t)})$ for $M_i^{(t)}$ times, compute player $i$'s average payoff $u_i^{(t)}(a)$
6:     For all $b \in \mathcal{A}_i$, set $\hat{\theta}_i^{(t+1)}(\cdot|b) \propto \exp\left(\eta_{t,i}^b \sum_{\tau=1}^t u_i^{(\tau)}(\cdot)\theta_i^{(\tau)}(b)\right)$
7:     Find $\theta_i^{(t+1)} \in \Delta(\mathcal{A}_i)$ such that $\theta_i^{(t+1)}(a) = \sum_{b\in\mathcal{A}_i} \hat{\theta}_i^{(t+1)}(a|b)\theta_i^{(t+1)}(b)$
8: For all $t \in [T]$ and $i \in [N]$, eliminate all actions in $\theta_i^{(t)}$ with probability smaller than $p$, then renormalize the vector to simplex as $\bar{\theta}_i^{(t)}$
9: **output:** $\left(\sum_{t=1}^T \otimes_{i=1}^n \bar{\theta}_i^{(t)}\right)/T$

---

## 5 LEARNING RATIONALIZABLE CORRELATED EQUILIBRIUM

In order to extend our results on $\epsilon$-CCE to $\epsilon$-CE, a natural approach would be augmenting Algorithm 2 with the celebrated Blum-Mansour reduction (Blum & Mansour, 2007) from swap-regret to external regret. In this reduction, one maintains $A$ instances of a no-regret algorithm $\{\text{Alg}_1, \cdots, \text{Alg}_A\}$. In iteration $t$, the player would stack the recommendations of the $A$ algorithms as a matrix, denoted by $\hat{\theta}^{(t)} \in \mathbb{R}^{A \times A}$, and compute its eigenvector $\theta^{(t)}$ as the randomized strategy in round $t$. After observing the actual payoff vector $u^{(t)}$, it will pass the weighted payoff vector $\theta^{(t)}(a)u^{(t)}$ to algorithm $\text{Alg}_a$ for each $a$. In this section, we focus on a fixed player $i$, and omit the subscript $i$ when it's clear from the context.

Applying this reduction to Algorithm 2 directly, however, would fail to preserve rationalizability since the weighted loss vector $\theta^{(t)}(a)u^{(t)}$ admit a smaller utility gap $\theta^{(t)}(a)\Delta$. Specifically, consider an action $b$ dominated by a mixed strategy $x$. In the payoff estimate of instance $a$,

$$\sum_{\tau=1}^t \theta^{(\tau)}(a)\left(u^{(\tau)}(b) - u^{(\tau)}(x)\right) \gtrsim \Delta \sum_{\tau=1}^t \theta^{(\tau)}(a) - \sqrt{\sum_{\tau=1}^t \frac{1}{M^{(\tau)}}} \not\geq 0, \qquad (3)$$

which means that we cannot guarantee the elimination of IDAs every round as in Eq (2).

In Algorithm 3, we address this by making $\sum_{\tau=1}^t \theta^{(\tau)}(a)$ play the role as $t$, tracking the progress of each no-regret instance separately. In time step $t$, we will compute the average payoff vector $u^{(t)}$ based on $M^{(t)}$ samples; then as in the Blum-Mansour reduction, we will update the $A$ instances of Hedge with weighted payoffs $\theta^{(t)}(a)u^{(t)}$ and will use the eigenvector of $\hat{\theta}$ as the strategy for the next round. The key detail here is our choice of parameters, which adapts to the past strategies $\{\theta^{(\tau)}\}_{\tau=1}^t$:

$$M_i^{(t)} := \left\lceil \max_a \frac{64\theta_i^{(t)}(a)}{\Delta^2 \cdot \sum_{\tau=1}^t \theta_i^{(\tau)}(a)} \right\rceil, \eta_{t,i}^a := \max\left\{\frac{2\ln(1/p)}{\Delta \sum_{\tau=1}^t \theta_i^{(\tau)}(a)}, \sqrt{\frac{A\ln A}{t}}\right\}, p = \frac{\min\{\epsilon, \Delta\}}{8AN}. \quad (4)$$

Compared to Eq (1), we are essentially replacing $t$ with an adaptive $\sum_{\tau=1}^t \theta^{(\tau)}(a)$. We can now improve (3) to

$$\sum_{\tau=1}^t \theta^{(\tau)}(a)\left(u^{(\tau)}(b) - u^{(\tau)}(x)\right) \gtrsim \Delta \sum_{\tau=1}^t \theta^{(\tau)}(a) - \sqrt{\sum_{\tau=1}^t \frac{\theta^{(\tau)}(a)^2}{M^{(\tau)}}} \gtrsim \Delta \sum_{\tau=1}^t \theta^{(\tau)}(a). \tag{5}$$

This together with our choice of $\eta_t^a$ allows us to ensure the rationalizability of every iterate. The full algorithm is presented in Algorithm 3.

We proceed to our theoretical guarantee for Algorithm 3. The analysis framework is largely similar to that of Algorithm 2. Our choice of $M_i^{(t)}$ is sufficient to ensure $\Delta$-rationalizability via Azuma-Hoeffding inequality, while swap-regret analysis of the algorithm proves that the average (clipped) strategy is indeed an $\epsilon$-CE. The full proof is deferred to Appendix D.

**Theorem 12.** *With parameters in Eq. (4), after $T = \widetilde{O}\left(\frac{A}{\epsilon^2} + \frac{A}{\Delta^2}\right)$ rounds, with probability $1 - 3\delta$, the output strategy of Algorithm 3 is a $\Delta$-rationalizable $\epsilon$-CE. The total sample complexity is $\widetilde{O}\left(\frac{LNA}{\Delta^2} + \frac{NA^2}{\min\{\Delta^2, \epsilon^2\}}\right)$.*

---
**Algorithm 4** Rationalizable $\epsilon$-CCE via Black-box Reduction

---
1: $(a_1^\star, \cdots, a_N^\star) \leftarrow$ `Algorithm 1`
2: For all $i \in [N]$, initialize $\mathcal{A}_i^{(1)} \leftarrow \{a_i^\star\}$
3: **for** $t = 1, 2, \ldots$ **do**
4:      Find an $\epsilon'$-CCE $\Pi$ with black-box algorithm $\mathcal{O}$ in the sub-game $\Pi_{i \in [N]} \mathcal{A}_i^{(t)}$
5:      $\forall i \in [N], a_i' \in \mathcal{A}_i$, evaluate $u_i(a_i', \Pi_{-i})$ for $M$ times and compute average $\hat{u}_i(a_i', \Pi_{-i})$
6:      **for** $i \in [N]$ **do**
7:          Let $a_i' \leftarrow \arg\max_{a \in \mathcal{A}_i} \hat{u}_i(a, \Pi_{-i})$          // Computing the empirical best response
8:          $\mathcal{A}_i^{(t+1)} \leftarrow \mathcal{A}_i^{(t)} \cup \{a_i'\}$
9:      **if** $\mathcal{A}_i^{(t)} = \mathcal{A}_i^{(t+1)}$ for all $i \in [N]$ **then**
10:          **return** $\Pi$

---

Compared to Theorem 6, our second term has an additional $A$ factor, which is quite reasonable considering that algorithms for learning $\epsilon$-CE take $\widetilde{O}(A^2\epsilon^{-2})$ samples, also $A$-times larger than the $\epsilon$-CCE rate.

## 6 Reduction-based Algorithms

While Algorithm 2 and 3 make use of one specific no-regret algorithm, namely Hedge (Exponential Weights), in this section, we show that arbitrary algorithms for finding CCE/CE can be augmented to find rationalizable CCE/CE. The sample complexity obtained via this reduction is comparable with those of Algorithm 2 and 3 when $L = \Theta(NA)$, but slightly worse when $L \ll NA$. Moreover, this black-box approach would enable us to derive algorithms for rationalizable equilibria with more desirable qualities, such as last-iterate convergence, when using equilibria-finding algorithms with these properties.

Suppose that we are given a black-box algorithm $\mathcal{O}$ that finds $\epsilon$-CCE in arbitrary games. We can then use this algorithm in the following "support expansion" manner. We start with a subgame of only rationalizable actions, which can be identified efficiently with Algorithm 1, and call $\mathcal{O}$ to find an $\epsilon$-CCE $\Pi$ for the subgame. Next, we check for each $i \in [N]$ if the best response to $\Pi_{-i}$ is contained in $\mathcal{A}_i$. If not, this means that the subgame's $\epsilon$-CCE may not be an $\epsilon$-CCE for the full game; in this case, the best response to $\Pi_{-i}$ would be a rationalizable action that we can safely include into the action set. On the other hand, if the best response falls in $\mathcal{A}_i$ for all $i$, we can conclude that $\Pi$ is also an $\epsilon$-CCE for the original game. The details are given by Algorithm 4, and our main theoretical guarantee is the following.

**Theorem 13.** *Algorithm 4 outputs a $\Delta$-rationalizable $\epsilon$-CCE with high probability, using at most $NA$ calls to the black-box CCE algorithm and $\widetilde{O}\left(\frac{N^2A^2}{\min\{\epsilon^2, \Delta^2\}}\right)$ additional samples.*

Using similar algorithmic techniques, we can develop a reduction scheme for rationalizable $\epsilon$-CE. The detailed description for this algorithm is deferred to Appendix E. Here we only state its main theoretical guarantee.

**Theorem 14.** *There exists an algorithm that outputs a $\Delta$-rationalizable $\epsilon$-CE with high probability, using at most $NA$ calls to a black-box CE algorithm and $\widetilde{O}\left(\frac{N^2A^3}{\min\{\epsilon^2, \Delta^2\}}\right)$ additional samples.*

## 7 Conclusion

In this paper, we consider two tasks: (1) learning rationalizable action profiles; (2) learning rationalizable equilibria. For task 1, we propose a conceptually simple algorithm whose sample complexity is significantly better than prior work (Wu et al., 2021). For task 2, we develop the first provably efficient algorithms for learning $\epsilon$-CE and $\epsilon$-CCE that are also rationalizable. Our algorithms are computationally efficient, enjoy sample complexity that scales polynomially with the number of players and are able to avoid iteratively dominated actions completely. Our results rely on several new techniques which might be of independent interests to the community. There remains a gap between our sample complexity upper bounds and the available lower bounds for both tasks, closing which is an important future research problem.

## ACKNOWLEDGEMENTS

This work is supported by Office of Naval Research N00014-22-1-2253. Dingwen Kong is partially supported by the elite undergraduate training program of School of Mathematical Sciences in Peking University.

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

## A   Further Details on Rationalizability

### A.1   Equivalence of Never-best-response and strict dominance

It is known that for finite normal form games, rationalizable actions are given by iterated elimination of never-best-response actions, which is in fact equivalent to the iterative elimination of strictly dominated actions (Osborne & Rubinstein, 1994, Lemma 60.1). Here, for completeness, we include a proof that the iterative elimination of of actions that are never $\Delta$-best-response gives the same definition as Definition 1. Notice that it suffices to show that for every subgame, the set of never $\Delta$-best response actions and the set of $\Delta$-dominated actions are the same.

**Proposition A.1.** *Suppose that an action* $a \in \mathcal{A}_i$ *is never a* $\Delta$-*best response,* i.e. $\forall \Pi_{-i} \in \Delta(\prod_{j \neq i} \mathcal{A}_i)$, $\exists u \in \Delta(\mathcal{A}_i)$ *such that*

$$u_i(a, \Pi_{-i}) \leq u_i(u, \Pi_{-1}) - \Delta.$$

*Then* $a$ *is also* $\Delta$-*dominated,* i.e. $\exists u \in \Delta(\mathcal{A}_i)$, $\forall \Pi_{-i} \in \Delta(\prod_{j \neq i} \mathcal{A}_i)$

$$u_i(a, \Pi_{-i}) \leq u_i(u, \Pi_{-1}) - \Delta.$$

*Proof.* That $a$ is never a $\Delta$-best response is equivalent to

$$\min_{\Pi_{-1}} \max_u \{u_i(a, \Pi_{-i}) - u_i(u, \Pi_{-1})\} \leq -\Delta.$$

That $a$ is $\Delta$-dominated is equivalent to

$$\max_u \min_{\Pi_{-1}} \{u_i(a, \Pi_{-i}) - u_i(u, \Pi_{-1})\} \leq -\Delta.$$

Equivalence immediately follows from von Neumman's minimax theorem. □

### A.2   Proof of Proposition 1

*Proof.* We prove this inductively with the following hypothesis:

$$\forall l \geq 1, \forall i \in [N], \qquad \sum_{a \in \mathcal{A}_i} x_i^*(a) \cdot \mathbb{1}[a \in E_l] \leq \frac{2l\epsilon}{\Delta}.$$

**Base case:**   By the definition of $\epsilon$-NE, $\forall i \in [N], \forall x' \in \Delta(\mathcal{A}_i)$,

$$u_i(x_i^*, x_{-i}^*) \geq u_i(x', x_{-i}^*) - \epsilon.$$

Note that if $\widetilde{a} \in E_1 \cap \mathcal{A}_i$, $\exists x \in \Delta(\mathcal{A}_i)$ such that $\forall a_{-i}$,

$$u_i(\widetilde{a}, a_{-i}) \leq u_i(x, a_{-i}) - \Delta.$$

Therefore if we choose

$$x' := x_i^* - \sum_{a \in \mathcal{A}_i} \mathbb{1}[a \in E_1] x_i^*(a) \mathbf{e}_a + \sum_{a \in \mathcal{A}_i} \mathbb{1}[a \in E_1] x_i^*(a) \cdot x(a),$$

that is if we play the dominating strategy instead of the dominated action in $x_i^*$, then

$$u_i(x', x_{-i}^*) \geq u_i(x_i^*, x_{-i}^*) + \sum_{a \in \mathcal{A}_i} x_i^*(a) \cdot \mathbb{1}[a \in E_1]\Delta.$$

It follows that

$$\sum_{a \in \mathcal{A}_i} x_i^*(a) \cdot \mathbb{1}[a \in E_1] \leq \frac{\epsilon}{\Delta}.$$

**Induction step:** By the induction hypothesis, $\forall i \in [N]$,

$$\sum_{a \in \mathcal{A}_i} x_i^*(a) \cdot \mathbb{1}[a \in E_l] \le \frac{2l\epsilon}{\Delta}.$$

Now consider

$$\widetilde{x}_i := \frac{x_i^* - \sum_{a \in \mathcal{A}_i} \mathbb{1}[a \in E_l] \cdot x_i^*(a)\mathbf{e}_a}{1 - \sum_{a \in \mathcal{A}_i} \mathbb{1}[a \in E_l] \cdot x_i^*(a)}, \qquad (\forall i \in [N])$$

which is supported on actions on in $E_l$. The induction hypothesis implies $\|\widetilde{x}_i - x_i^*\|_1 \le 6l\epsilon/\Delta$. Therefore $\forall i \in [N], \forall a \in \mathcal{A}_i$,

$$\left| u_i(a, \widetilde{x}_{-i}) - u_i(a, x_{-i}^*) \right| \le \frac{6Nl\epsilon}{\Delta}.$$

Now if $\widetilde{a} \in (E_{l+1} \setminus E_l) \cap \mathcal{A}_i$, since $\widetilde{x}_{-i}$ is not supported on $E_l$, $\exists x \in \Delta(\mathcal{A}_i)$ such that

$$u_i(\widetilde{a}, \widetilde{x}_{-i}) \le u_i(x, \widetilde{x}_{-i}) - \Delta.$$

It follows that

$$u_i(\widetilde{a}, x_{-i}^*) \le u_i(x, x_{-i}^*) - \Delta + \frac{12Nl\epsilon}{\Delta} \le u_i(x, x_{-i}^*) - \frac{\Delta}{2}.$$

Using the same arguments as in the base case,

$$\sum_{a \in \mathcal{A}_i} x_i^*(a) \cdot \mathbb{1}[a \in E_{l+1} \setminus E_l] \le \frac{\epsilon}{\Delta - \frac{12Nl\epsilon}{\Delta}} \le \frac{2\epsilon}{\Delta}.$$

It follows that $\forall i \in [N]$,

$$\sum_{a \in \mathcal{A}_i} x_i^*(a) \cdot \mathbb{1}[a \in E_{l+1}] \le \frac{2(l+1)\epsilon}{\Delta}.$$

The statement is thus proved via induction on $l$. $\qquad \square$

## B  FIND ONE RATIONALIZABLE ACTION PROFILE

### B.1  PROOF OF PROPOSITION 2

*Proof.* Consider the following $N$-player game denoted by $G_0$ with action set $[A]$:

$$u_i(\cdot) = 0 \qquad\qquad (1 \le i \le N-1)$$
$$u_N(a_N) = \Delta \cdot \mathbb{1}[a_N > 1].$$

Specifically, a payoff with mean $u$ is realized by a skewed Rademacher random variable with $\frac{1+u}{2}$ probability on $+1$ and $\frac{1-u}{2}$ on $-1$. In game $G_0$, clearly for player $N$, the action 1 is $\Delta$-dominated. However, consider the following game, denoted by $G_{a^*}$ (where $a^* \in [A]^{N-1}$)

$$u_i(\cdot) = 0, \qquad\qquad (1 \le i \le N-1)$$
$$u_N(a_N) = \Delta, \qquad\qquad (a_N > 1)$$
$$u_N(1, a_{-N}) = 2\Delta \cdot \mathbb{1}[a_{-N} = a^*].$$

It can be seen that in game $G_{a^*}$, for player $N$, the action 1 is *not* dominated or iteratively strictly dominated. Therefore, suppose that an algorithm $\mathcal{O}$ is able to determine whether an action is rationalizable (i.e. not iteratively strictly dominated) with 0.9 accuracy, then its output needs to be False with at least 0.9 probability in game $G_0$, but True with at least 0.9 probability in game $G_{a^*}$. By Pinsker's inequality,

$$\mathrm{KL}(\mathcal{O}(G_0)\|\mathcal{O}(G_{a^*})) \ge 2 \cdot 0.8^2 > 1,$$

where we used $\mathcal{O}(G)$ to denote the trajectory generated by running algorithm $\mathcal{O}$ on game $G$. Meanwhile, notice that $G_0$ and $G_{a^*}$ is different only when the first $N-1$ players play $a^*$. Denote

the number of times where the first $N-1$ players play $a^*$ by $n(a^*)$. Using the chain rule of KL-divergence,

$$\mathrm{KL}(\mathcal{O}(G_0)||\mathcal{O}(G_{a^*})) \leq \mathbb{E}_{G_0}\left[n(a^*)\right] \cdot \mathrm{KL}\left(\mathrm{Ber}\left(\frac{1}{2}\right)\middle\|\mathrm{Ber}\left(\frac{1+2\Delta}{2}\right)\right)$$

$$\overset{(a)}{\leq} \mathbb{E}_{G_0}\left[n(a^*)\right] \cdot \frac{1}{\frac{1-2\Delta}{2}} \cdot (2\Delta)^2$$

$$\overset{(b)}{\leq} 10\Delta^2\mathbb{E}_{G_0}\left[n(a^*)\right].$$

Here $(a)$ follows from reverse Pinsker's inequality (see *e.g.* Binette (2019)), while $(b)$ uses the fact that $\Delta < 0.1$. This means that for any $a^* \in [A]^{N-1}$,

$$\mathbb{E}_{G_0}\left[n(a^*)\right] \geq \frac{1}{10\Delta^2}.$$

It follows that the expected number of samples when running $\mathcal{O}$ on $G_0$ is at least

$$\mathbb{E}_{G_0}\left[\sum_{a^* \in [A]^{N-1}} n(a^*)\right] \geq \frac{A^{N-1}}{10\Delta^2}.$$

$\square$

## B.2 PROOF OF THEOREM 3

*Proof.* We first present the concentration bound. For $l \in [L]$, $i \in [N]$, and $a \in \mathcal{A}_i$, by Hoeffding's inequality we have that with probability at least $1 - \frac{\delta}{LNA}$,

$$\left|u_i(a, a_{-i}^{(l-1)}) - \hat{u}_i(a, a_{-i}^{(l-1)})\right| \leq \sqrt{\frac{4\ln(ANL/\delta)}{M}} \leq \frac{\Delta}{4}.$$

Therefore by a union bound we have that with probability at least $1 - \delta$, for all $l \in [L]$, $i \in [N]$, and $a \in \mathcal{A}_i$,

$$\left|u_i(a, a_{-i}^{(l-1)}) - \hat{u}_i(a, a_{-i}^{(l-1)})\right| \leq \frac{\Delta}{4}.$$

We condition on this event for the rest of the proof.

We use induction on $l$ to prove that for all $l \in [L] \cup \{0\}$, $(a_1^{(l)}, \cdots, a_N^{(l)})$ can survive at least $l$ rounds of IDE. The base case for $l = 0$ directly holds. Now we assume that the case for $1, 2, \ldots, l-1$ holds and consider the case of $l$.

For any $i \in [N]$, we show that $a_i^{(l)}$ can survive at least $l$ rounds of IDE. Recall that $a_i^{(l)}$ is the empirical best response, *i.e.*

$$a_i^{(l)} = \arg\max_{a \in \mathcal{A}_i} \hat{u}_i(a, a_{-i}^{(l-1)}).$$

For any mixed strategy $x_i \in \Delta(\mathcal{A}_i)$, we have that

$$u_i(a_i^{(l)}, a_{-i}^{(l-1)}) - u_i(x_i, a_{-i}^{(l-1)})$$

$$\geq \hat{u}_i(a_i^{(l)}, a_{-i}^{(l-1)}) - \hat{u}_i(x_i, a_{-i}^{(l-1)}) - \left|u_i(a_i^{(l)}, a_{-i}^{(l-1)}) - \hat{u}_i(a_i^{(l)}, a_{-i}^{(l-1)})\right| - \left|u_i(x_i, a_{-i}^{(l-1)}) - \hat{u}_i(x_i, a_{-i}^{(l-1)})\right|$$

$$\geq 0 - \frac{\Delta}{4} - \frac{\Delta}{4} = -\frac{\Delta}{2}.$$

Since actions in $a_{-i}^{(l-1)}$ can survive at least $l-1$ rounds of $\Delta$-IDE, $a_i^{(l)}$ cannot be $\Delta$-dominated by $x_i$ in rounds $1, \cdots, l$. Since $x_i$ can be arbitrarily chosen, $a_i^{(l)}$ can survive at least $l$ rounds of $\Delta$-IDE. We can now ensure that the output $(a_1^{(L)}, \cdots, a_N^{(L)})$ survives $L$ rounds of $\Delta$-IDE, which is equivalent to $\Delta$-rationalizability (see Definition 1).

The total number of samples used is

$$LNA \cdot M = \widetilde{O}\left(\frac{LNA}{\Delta^2}\right).$$

$\square$

### B.3  PROOF OF THEOREM 4

*Proof.* Without loss of generality, assume that $\Delta < 0.1$. Consider the following instance where $\mathcal{A}_1 = \cdots = \mathcal{A}_N = [A]$:

$$u_i(a_i) = \Delta \cdot \mathbb{1}[a_i = 1], \qquad\qquad (i \neq j)$$

$$u_j(a_j, a_{-j}) = \begin{cases} \Delta \cdot \mathbb{1}[a_j = 1] & (a_{-j} \neq \{1\}^{N-1}) \\ \Delta \cdot \mathbb{1}[a_j = 1] + 2\Delta \cdot \mathbb{1}[a_j = a] & (a_{-j} = \{1\}^{N-1}) \end{cases}.$$

Denote this instance by $G_{j,a}$. Additionally, define the following instance $G_0$:

$$u_i(a_i) = \Delta \cdot \mathbb{1}[a_i = 1]. \qquad\qquad (\forall i \in [N])$$

As before, a payoff with expectation $u$ is realized as a random variable with distribution $2\mathrm{Ber}(\frac{1+u}{2})-1$. It can be seen that the only difference between $G_0$ and $G_{j,a}$ lies in $u_j(a, \{1\}^{N-1})$. By the KL-divergence chain rule, for any algorithm $\mathcal{O}$,

$$KL\left(\mathcal{O}(G_0)\| \mathcal{O}(G_{j,a})\right) \leq 10\Delta^2 \cdot \mathbb{E}_{G_0}\left[n(a_j = a, a_{-j} = \{1\}^{N-1})\right],$$

where $n(a_j = a, a_{-j} = \{1\}^{N-1})$ denotes the number of times the action profile $(a, 1^{N-1})$ is played. Note that in $G_0$, the only action profile surviving two rounds of $\Delta$-IDE is $(1, \cdots, 1)$, while in $G_{j,a}$, the only rationalizable action profile is $(\underbrace{1, \cdots, 1}_{j-1}, a, 1, \cdots, 1)$. To guarantee 0.9 accuracy, by Pinsker's inequality,

$$\mathrm{KL}\left(\mathcal{O}(G_0)\|\mathcal{O}(G_{j,a})\right) \geq \frac{1}{2}\left|\mathcal{O}(G_0) - \mathcal{O}(G_{j,a})\right|^2 > 1.$$

It follows that $\forall j \in [N], a > 1$,

$$\mathbb{E}_{G_0}\left[n(a_j = a, a_{-j} = \{1\}^{N-1})\right] \geq \frac{1}{10\Delta^2}.$$

Thus the total expected sample complexity is at least

$$\sum_{a>1, j\in[N]} \mathbb{E}_{G_0}\left[n(a_j = a, a_{-j} = \{1\}^{N-1})\right] \geq \frac{N(A-1)}{10\Delta^2}.$$

$\square$

## C  OMITTED PROOFS IN SECTION 4

We start our analysis by bounding the sampling noise. For player $i \in [N]$, action $a_i \in \mathcal{A}_i$, and $\tau \in [T]$, we denote the sampling noise as

$$\xi_i^{(\tau)}(a_i) := u_i^{(\tau)}(a_i) - u_i(a_i, \theta_{-i}^{(\tau)}).$$

We have the following lemma.

**Lemma C.1.** *Let $\Omega_1$ denote the event that for all $t \in [T]$, $i \in [N]$, and $a_i \in \mathcal{A}_i$,*

$$\left|\sum_{\tau=1}^{t} \xi_i^{(\tau)}(a_i)\right| \leq 2\sqrt{\ln(ANT/\delta)\sum_{\tau=1}^{t}\frac{1}{M_\tau}}.$$

*Then $\Pr[\Omega_1] \geq 1 - \delta$.*

*Proof.* Note that $\sum_{\tau=1}^{t} \xi_i^{(\tau)}(a_i)$ can be written as the sum of $\sum_{\tau=1}^{t} M_\tau$ mean-zero bounded terms. By Azuma-Hoeffding inequality, with probability at least $1 - \frac{\delta}{ANT}$, for a fixed $i \in [N]$, $t \in [T]$, $a_i \in \mathcal{A}_i$,

$$\left|\sum_{\tau=1}^{t} \xi_i^{(\tau)}(a_i)\right| \leq 2\sqrt{\ln(ANT/\delta)\sum_{\tau=1}^{t} M_\tau \cdot \left(\frac{1}{M_\tau}\right)^2}. \qquad (6)$$

A union bound over $i \in [N]$, $t \in [T]$, $a_i \in \mathcal{A}_i$ proves the statement. $\square$

**Lemma C.2.** *With probability at least $1 - 2\delta$, for all $t \in [T]$ and all $i \in [N]$, $a_i \in \mathcal{A}_i \cap E_L$,*

$$\theta_i^{(t)}(a_i) \leq p.$$

*Proof.* We condition on the event $\Omega_1$ defined in Lemma C.1 and the success of Algorithm 1. We prove the claim by induction in $t$. The base case for $t = 1$ holds directly by initialization. Now we assume the case for $1, 2, \ldots, t$ holds and consider the case of $t + 1$.

Consider a fixed player $i \in [N]$ and iteratively dominated action $a_i \in \mathcal{A}_i \cap E_L$. By definition there exists a mixed strategy $x_i$ such that for all $a_{-i} \cap E_L = \emptyset$,

$$u_i(x_i, a_{-i}) \geq u_i(a_i, a_{-i}) + \Delta.$$

Therefore for $\tau \in [t]$, by the induction hypothesis for $\tau$,

$$u_i(x_i, \theta_{-i}^{(\tau)}) \geq u_i(a_i, \theta_{-i}^{(\tau)}) + (1 - ANp) \cdot \Delta - ANp$$
$$\geq u_i(a_i, \theta_{-i}^{(\tau)}) + \Delta/2. \tag{7}$$

Consequently,

$$\sum_{\tau=1}^{t}(u_i^{(\tau)}(x_i) - u_i^{(\tau)}(a_i))$$

$$\geq \sum_{\tau=1}^{t}(u_i(x_i, \theta_{-i}^{(\tau)}) - u_i(a_i, \theta_{-i}^{(\tau)})) - 4 \cdot \sqrt{\ln(ANT/\delta) \sum_{\tau=1}^{t} \frac{1}{M_\tau}} \qquad \text{(By (6))}$$

$$\geq \frac{t\Delta}{2} - 4 \cdot \sqrt{\ln(ANT/\delta) \sum_{\tau=1}^{t} \frac{1}{M_\tau}} \qquad \text{(By (7))}$$

$$\geq \frac{t\Delta}{4}.$$

Therefore by our choice of learning rate,

$$\theta_i^{(t+1)}(a_i) \leq \exp\left(-\eta_t \cdot \sum_{\tau=1}^{t} \left(u_i^{(\tau)}(x_i) - u_i^{(\tau)}(a_i)\right)\right)$$

$$\leq \exp\left(-\frac{4\ln(1/p)}{\Delta t} \cdot \frac{\Delta t}{4}\right) = p.$$

Therefore

$$\theta_i^{(t+1)}(a_i) \leq p$$

as desired. $\qquad \square$

Now we turn to the $\epsilon$-CCE guarantee. For a player $i \in [N]$, recall that the regret is defined as

$$\text{Regret}_T^i = \max_{\theta \in \Delta(\mathcal{A}_i)} \sum_{t=1}^{T} \langle u_i^{(t)}, \theta - \theta_i^{(t)} \rangle.$$

**Lemma C.3.** *The regret can be bounded as*

$$\text{Regret}_T^i \leq O\left(\sqrt{\ln A \cdot T} + \frac{\ln(1/p)\ln T}{\Delta}\right).$$

*Proof.* Note that apart from the choice of $\theta^{(1)}$, we are exactly running FTRL with learning rates

$$\eta_t = \max\left\{\sqrt{\ln A/t}, \frac{4\ln(1/p)}{\Delta t}\right\},$$

which are monotonically decreasing. Therefore following the standard analysis of FTRL (see, *e.g.*, Orabona (2019, Corollary 7.9)), we have

$$\max_{\theta \in \Delta(\mathcal{A}_i)} \sum_{t=1}^{T} \langle u_i^{(t)}, \theta - \theta_i^{(t)} \rangle \leq 2 + \frac{\ln A}{\eta_T} + \frac{1}{2} \sum_{t=1}^{T} \eta_t$$

$$\leq 2 + \sqrt{\ln A \cdot T} + \frac{1}{2} \sum_{t=1}^{T} \left( \sqrt{\frac{\ln A}{t}} + \frac{4 \ln(1/p)}{\Delta t} \right)$$

$$= O\left( \sqrt{\ln A \cdot T} + \frac{\ln(1/p) \ln T}{\Delta} \right).$$

$\square$

However, this form of regret cannot directly imply approximate CCE. We define the following expected version regret

$$\text{Regret}_T^{i,\star} = \max_{\theta \in \Delta(\mathcal{A}_i)} \sum_{t=1}^{T} \langle u_i(\cdot, \theta_{-i}^{(t)}), \theta - \theta_i^{(t)} \rangle.$$

The next lemma bound the difference between these two types of regret

**Lemma C.4.** *The following event $\Omega_2$ holds with probability at least $1 - \delta$: for all $i \in [N]$*

$$\left| \text{Regret}_T^{i,\star} - \text{Regret}_T^{i} \right| \leq O\left( \sqrt{T \cdot \ln(NA/\delta)} \right).$$

*Proof.* We denote

$$\Theta_i := \{ \mathbf{e}_1, \mathbf{e}_2, \ldots, \mathbf{e}_{|\mathcal{A}_i|} \}$$

Therefore we have

$$\left| \text{Regret}_T^{i,\star} - \text{Regret}_T^{i} \right|$$

$$= \left| \max_{\theta \in \Delta(\mathcal{A}_i)} \sum_{t=1}^{T} \langle u_i(\cdot, \theta_{-i}^{(t)}), \theta - \theta_i^{(t)} \rangle - \max_{\theta \in \Delta(\mathcal{A}_i)} \sum_{t=1}^{T} \langle u_i^{(t)}, \theta - \theta_i^{(t)} \rangle \right|$$

$$= \left| \max_{\theta \in \Theta_i} \sum_{t=1}^{T} \langle u_i(\cdot, \theta_{-i}^{(t)}), \theta - \theta_i^{(t)} \rangle - \max_{\theta \in \Theta_i} \sum_{t=1}^{T} \langle u_i^{(t)}, \theta - \theta_i^{(t)} \rangle \right|$$

$$= \max_{\theta \in \Theta_i} \left| \sum_{t=1}^{T} \langle u_i(\cdot, \theta_{-i}^{(t)}), \theta - \theta_i^{(t)} \rangle - \sum_{t=1}^{T} \langle u_i^{(t)}, \theta - \theta_i^{(t)} \rangle \right|$$

$$= \max_{\theta \in \Theta_i} \left| \sum_{t=1}^{T} \langle u_i(\cdot, \theta_{-i}^{(t)}) - u_i^{(t)}, \theta - \theta_i^{(t)} \rangle \right|$$

Note that $\langle u_i(\cdot, \theta_{-i}^{(t)}) - u_i^{(t)}, \theta - \theta_i^{(t)} \rangle$ is a bounded martingale difference sequence. By Azuma-Hoeffding's inequality, for a fixed $\theta \in \Theta_i$, with probability at least $1 - \frac{\delta}{AN}$,

$$\left| \sum_{t=1}^{T} \langle u_i(\cdot, \theta_{-i}^{(t)}) - u_i^{(t)}, \theta - \theta_i^{(t)} \rangle \right| \leq O\left( \sqrt{T \cdot \ln(NA/\delta)} \right)$$

Thus we complete the proof by a union bound. $\square$

*Proof of Theorem 6.* We condition on event $\Omega_1$ defined Lemma C.1, event $\Omega_2$ defined in Lemma C.4, and the success of Algorithm 1.

**Coarse Correlated Equilibria.** By Lemma C.3 and Lemma C.4 we know that for all $i \in [N]$,

$$\text{Regret}_T^{i,\star} \leq O\left( \sqrt{\ln A \cdot T} + \frac{\ln(1/p) \ln T}{\Delta} + \sqrt{T \cdot \ln(NA/\delta)} \right).$$

Therefore choosing

$$T = \Theta \left( \frac{\ln(NA/\delta)}{\epsilon^2} + \frac{\ln^2(NA/\Delta\epsilon\delta)}{\Delta\epsilon} \right)$$

will guarantee that $\mathrm{Regret}_T^{i,\star}$ is at most $\epsilon T/2$ for all $i \in [N]$. In this case the average strategy $(\sum_{t=1}^T \otimes_{i=1}^N \theta_i^{(t)})/T$ would be an $(\epsilon/2)$-CCE.

Finally, in the clipping step, $\|\bar{\theta}_i^{(t)} - \theta_i^{(t)}\|_1 \le 2pA \le \frac{\epsilon}{4N}$ for all $i \in [N]$, $t \in [T]$. Thus for all $t \in [T]$, we have $\| \otimes_{i=1}^n \bar{\theta}_i^{(t)} - \otimes_{i=1}^n \theta_i^{(t)} \|_1 \le \frac{\epsilon}{4}$, which further implies

$$\left\| (\sum_{t=1}^T \otimes_{i=1}^n \bar{\theta}_i^{(t)})/T - (\sum_{t=1}^T \otimes_{i=1}^n \theta_i^{(t)})/T \right\|_1 \le \frac{\epsilon}{4}.$$

Therefore the output strategy $\Pi = (\sum_{t=1}^T \otimes_{i=1}^N \bar{\theta}_i^{(t)})/T$ is an $\epsilon$-CCE.

**Rationalizability.** By Lemma C.2, if $a \in E_L \cap \mathcal{A}_i$, $\theta_i^{(t)}(a) \le p$ for all $t \in [T]$. It follows that $\bar{\theta}_i^{(t)}(a) = 0$, *i.e.*, the action would not be the support in the output strategy $\Pi = (\sum_{t=1}^T \otimes_{i=1}^N \bar{\theta}_i^{(t)})/T$.

**Sample complexity**. The total number of full-information queries is

$$\sum_{t=1}^T M_t \le T + \sum_{t=1}^T \frac{64\ln(ANT/\delta)}{\Delta^2 t}$$

$$\le T + \widetilde{O}\left( \frac{1}{\Delta^2} \right)$$

$$= \widetilde{O}\left( \frac{1}{\Delta^2} + \frac{1}{\epsilon^2} \right).$$

The total sample complexity for CCE learning would then be

$$NA \cdot \sum_{t=1}^T M_t = \widetilde{O}\left( \frac{NA}{\epsilon^2} + \frac{NA}{\Delta^2} \right).$$

Finally consider the cost of finding one IDE-surviving action profile ($\widetilde{O}\left( \frac{LNA}{\Delta^2} \right)$) and we get the claimed rate. $\qquad \square$

# D    OMITTED PROOFS IN SECTION 5

Similar to the CCE case we first bound the sampling noise. For action $a_i \in \mathcal{A}_i$, and $\tau \in [T]$, we denote the sampling noise as

$$\xi_i^{(\tau)}(a_i) := u_i^{(\tau)}(a_i) - u_i(a_i, \theta_{-i}^{(\tau)}).$$

In the CE case, we are interested in the weighted sum of noise $\sum_{\tau=1}^t \xi_i^{(\tau)}(a_i)\theta_i^{(\tau)}(b_i)$, which is bounded in the following lemma.

**Lemma D.1.** *The following event $\Omega_3$ holds with probability at least $1 - \delta$: for all $t \in [T]$, $i \in [N]$, and $a_i \in \mathcal{A}_i$,*

$$\left| \sum_{\tau=1}^t \xi_i^{(\tau)}(a_i)\theta_i^{(\tau)}(b_i) \right| \le \frac{\Delta}{4} \sum_{\tau=1}^t \theta_i^{(\tau)}(b_i).$$

*Proof.* Note that $\sum_{\tau=1}^t \xi_i^{(\tau)}(a_i)\theta_i^{(\tau)}(b_i)$ can be written as the sum of $\sum_{\tau=1}^t M_i^\tau$ mean-zero bounded terms. Precisely, there are $M_i^\tau$ terms bounded by $\frac{\theta_i^{(\tau)}(b_i)}{M_i^\tau}$. By the Azuma-Hoeffding inequality, we

have that with probability at least $1 - \frac{\delta}{A^2 NT}$,

$$
\left| \sum_{\tau=1}^{t} \xi_i^{(\tau)}(a_i) \theta_i^{(\tau)}(b_i) \right| \leq 2 \cdot \sqrt{\ln(ANT/\delta) \sum_{\tau=1}^{t} M_i^{\tau} \cdot \left( \frac{\theta_i^{(\tau)}(b_i)}{M_i^{\tau}} \right)^2}
$$

$$
= 2 \cdot \sqrt{\ln(ANT/\delta) \sum_{\tau=1}^{t} \frac{(\theta_i^{(\tau)}(b_i))^2}{M_i^{\tau}}}
$$

$$
\leq \frac{\Delta}{4} \cdot \sqrt{\sum_{\tau=1}^{t} \theta_i^{(\tau)}(b_i) \sum_{j=1}^{\tau} \theta_i^{(j)}(b_i)}
$$

$$
\leq \frac{\Delta}{4} \sum_{\tau=1}^{t} \theta_i^{(\tau)}(b_i)
$$

Therefore by a union bound we complete the proof. $\qquad \square$

**Lemma D.2.** *With probability at least $1 - 2\delta$, for all $t \in [T]$, all $i \in [N]$, and all $a_i \in \mathcal{A}_i \cap E_L$,*

$$
\theta_i^{(t)}(a_i) \leq p
$$

*Proof.* We condition on the event $\Omega_3$ defined in Lemma D.1 and the success of Algorithm 1. We prove the claim by induction in $t$. The base case for $t = 1$ holds directly by initialization. Now we assume the case for $1, 2, \ldots, t$ holds and consider the case of $t + 1$.

Consider a fixed player $i \in [N]$, an iteratively dominated action $a_i \in \mathcal{A}_i \cap E_L$, and an expert $b_i$. By definition there exists a mixed strategy $x_i$ such that for all $a_{-i} \cap E_L = \emptyset$,

$$
u_i(x_i, a_{-i}) \geq u_i(a_i, a_{-i}) + \Delta
$$

Therefore for $\tau \in [t]$, by induction hypothesis we have

$$
u_i(x_i, \theta_{-i}^{(\tau)}) \geq u_i(a_i, \theta_{-i}^{(\tau)}) + (1 - ANp) \cdot \Delta - ANp
$$

$$
\geq u_i(a_i, \theta_{-i}^{(\tau)}) + \Delta/2
$$

Thus we have

$$
\sum_{\tau=1}^{t} (u_i^{(\tau)}(x_i) - u_i^{(\tau)}(a_i)) \cdot \theta_i^{(\tau)}(b_i)
$$

$$
\geq \sum_{\tau=1}^{t} (u_i(x_i, \theta_{-i}^{(\tau)}) - u_i(a_i, \theta_{-i}^{(\tau)})) \cdot \theta_i^{(\tau)}(b_i) - \frac{\Delta}{4} \sum_{\tau=1}^{t} \theta_i^{(\tau)}(b_i)
$$

$$
\geq \frac{\Delta}{2} \sum_{\tau=1}^{t} \theta_i^{(\tau)}(b_i) - \frac{\Delta}{4} \sum_{\tau=1}^{t} \theta_i^{(\tau)}(b_i)
$$

$$
= \frac{\Delta}{4} \sum_{\tau=1}^{t} \theta_i^{(\tau)}(b_i)
$$

By our choice of learning rate,

$$
\hat{\theta}_i^{(t+1)}(a_i|b_i) \leq \exp \left( -\eta_{t,i}^{b_i} \cdot \sum_{\tau=1}^{t} \theta_i^{(\tau)}(b_i) \left( u_i^{(\tau)}(x_i) - u_i^{(\tau)}(a_i) \right) \right)
$$

$$
\leq \exp \left( -\frac{4\ln(1/p)}{\Delta \sum_{\tau=1}^{t} \theta_i^{(\tau)}(b)} \cdot \frac{\Delta}{4} \sum_{i=1}^{t} \theta_i^{(\tau)}(b) \right) = p.
$$

Therefore we conclude

$$
\theta_i^{(t+1)}(a_i) = \sum_{b_i \in \mathcal{A}_i} \hat{\theta}_i^{(t+1)}(a_i|b_i) \theta_i^{(t+1)}(b_i) \leq p
$$

$\qquad \square$

Now we turn to the $\epsilon$-CE guarantee. For a player $i \in [N]$, recall that the swap-regret is defined as

$$\text{SwapRegret}_T^i := \sup_{\phi: \mathcal{A}_i \to \mathcal{A}_i} \sum_{t=1}^{T} \sum_{b \in \mathcal{A}_i} \theta_i^{(t)}(b) u_i^{(t)}(\phi(b)) - \sum_{t=1}^{T} \left\langle \theta_i^{(t)}, u_i^{(t)} \right\rangle.$$

**Lemma D.3.** *For all $i \in [N]$, the swap-regret can be bounded as*

$$\text{SwapRegret}_T^i \leq O\left(\sqrt{A \ln(A)T} + \frac{A \ln(NAT/\Delta\epsilon)^2}{\Delta}\right).$$

*Proof.* For $i \in [N]$, recall that the regret for an expert $b \in \mathcal{A}_i$ is defined as

$$\text{Regret}_T^{i,b} := \max_{a \in \mathcal{A}_i} \sum_{t=1}^{T} \theta_i^{(t)}(b) u_i^{(t)}(a) - \sum_{t=1}^{T} \left\langle \hat{\theta}_i^{(t)}(\cdot|b), \theta_i^{(t)}(b) u_i^{(t)} \right\rangle.$$

Since $\theta_i^{(t)}(a) = \sum_{b \in \mathcal{A}_i} \hat{\theta}_i^{(t)}(a|b)\theta_i^{(t)}(b)$ for all $a$ and all $t > 1$,

$$\sum_{b \in \mathcal{A}_i} \text{Regret}_T^{i,b} = \sum_{b \in \mathcal{A}_i} \max_{a_b \in \mathcal{A}_i} \sum_{t=1}^{T} \theta_i^{(t)}(b) u_i^{(t)}(a_b) - \sum_{b \in \mathcal{A}_i} \sum_{t=1}^{T} \left\langle \hat{\theta}_i^{(t)}(\cdot|b)\theta_i^{(t)}(b), u_i^{(t)} \right\rangle$$

$$= \max_{\phi: \mathcal{A}_i \to \mathcal{A}_i} \sum_{b \in \mathcal{A}_i} \sum_{t=1}^{T} \theta_i^{(t)}(b) u_i^{(t)}(\phi(b)) - \sum_{t=1}^{T} \left\langle \sum_{b \in \mathcal{A}_i} \hat{\theta}_i^{(t)}(\cdot|b)\theta_i^{(t)}(b), u_i^{(t)} \right\rangle$$

$$\geq \max_{\phi: \mathcal{A}_i \to \mathcal{A}_i} \sum_{t=1}^{T} \sum_{b \in \mathcal{A}_i} \theta_i^{(t)}(b) u_i^{(t)}(\phi(b)) - \sum_{t=2}^{T} \left\langle \theta_i^{(t)}, u_i^{(t)} \right\rangle - 1 \geq \text{SwapRegret}_T^i - 1.$$

It now suffices to control the regret of each individual expert. For expert $b$, we are essentially running FTRL with learning rates

$$\eta_{t,i}^b := \max\left\{\frac{4\ln(1/p)}{\Delta \sum_{\tau=1}^{t} \theta_i^{(\tau)}(b)}, \frac{\sqrt{A \ln A}}{\sqrt{t}}\right\},$$

which are clearly monotonically decreasing. Therefore using standard analysis of FTRL (see, *e.g.*, Orabona (2019, Corollary 7.9)),

$$\text{Regret}_T^{i,b} \leq \frac{\ln A}{\eta_{T,i}^b} + \sum_{t=1}^{T} \eta_{t,i}^b \cdot \theta_i^{(t)}(b)^2$$

$$\leq \sqrt{\frac{T \ln A}{A}} + \sum_{t=1}^{T} \theta_i^{(t)}(b) \cdot \sqrt{\frac{A \ln A}{t}} + \frac{4\ln(1/p)}{\Delta} \cdot \sum_{t=1}^{T} \frac{\theta_i^{(t)}(b)}{\sum_{\tau=1}^{t} \theta_i^{(\tau)}(b)}$$

$$\leq \sqrt{\frac{T \ln A}{A}} + \sum_{t=1}^{T} \theta_i^{(t)}(b) \cdot \sqrt{\frac{A \ln A}{t}} + \frac{4\ln(1/p)}{\Delta}\left(1 + \ln\left(\frac{T}{p}\right)\right).$$

Here we used the fact that $\forall b \in \mathcal{A}_i$, $\theta_i^{(1)}(b) \geq p$, and

$$\sum_{t=1}^{T} \frac{\theta_i^{(t)}(b)}{\sum_{i=1}^{\tau} \theta_i^{(\tau)}(b)} \leq 1 + \int_{\theta_i^{(1)}(b)}^{\sum_{t=1}^{T} \theta_i^{(t)}(b)} \frac{\mathrm{d}s}{s} = 1 + \ln\left(\frac{\sum_{t=1}^{T} \theta_i^{(t)}(b)}{\theta_i^{(1)}(b)}\right)$$

$$\leq 1 + \ln\left(\frac{T}{p}\right).$$

Notice that $\sum_{b \in \mathcal{A}_i} \sum_{t=1}^{T} \theta_i^{(t)}(b) \cdot \sqrt{\frac{A \ln A}{t}} \leq O(\sqrt{A \ln(A)T})$. Therefore

$$\text{SwapRegret}_T^i \leq O(1) + \sum_{b \in \mathcal{A}_i} \text{Regret}_T^{i,b} \leq O\left(\sqrt{A \ln(A)T} + \frac{A \ln(NAT/\Delta\epsilon)^2}{\Delta}\right). \tag{8}$$

$\square$

Similar to the CCE case,, this form of regret can not directly imply approximate CE. We define the following expected version regret

$$\text{SwapRegret}_T^{i,\star} := \sup_{\phi:\mathcal{A}_i \to \mathcal{A}_i} \sum_{t=1}^T \left\langle \phi \circ \theta_i^{(t)}, u_i(\cdot, \theta_{-i}^{(t)}) \right\rangle - \sum_{t=1}^T \left\langle \theta_i^{(t)}, u_i(\cdot, \theta_{-i}^{(t)}) \right\rangle$$

The next lemma bound the difference between these two types of regret

**Lemma D.4.** *The following event $\Omega_4$ has probability at least $1 - \delta$: for all $i \in [N]$,*

$$\left| \text{SwapRegret}_T^{i,\star} - \text{SwapRegret}_T^i \right| \le O\left( \sqrt{AT \ln\left(\frac{AN}{\delta}\right)} \right).$$

*Proof.* Note that

$$\left| \text{SwapRegret}_T^{i,\star} - \text{SwapRegret}_T^i \right|$$

$$= \left| \sup_{\phi:\mathcal{A}_i \to \mathcal{A}_i} \sum_{t=1}^T \left\langle \phi \circ \theta_i^{(t)} - \theta_i^{(t)}, u_i(\cdot, \theta_{-i}^{(t)}) \right\rangle - \sup_{\phi:\mathcal{A}_i \to \mathcal{A}_i} \sum_{t=1}^T \left\langle \phi \circ \theta_i^{(t)} - \theta_i^{(t)}, u_i^{(t)} \right\rangle \right|$$

$$\le \sup_{\phi:\mathcal{A}_i \to \mathcal{A}_i} \left| \sum_{t=1}^T \left\langle \phi \circ \theta_i^{(t)} - \theta_i^{(t)}, u_i(\cdot, \theta_{-i}^{(t)}) - u_i^{(t)} \right\rangle \right|.$$

Notice that $\mathbb{E}[u_i^{(t)}] = u_i\left(\cdot, \theta_{-i}^{(t)}\right)$, and that $u_i^{(t)} \in [-1, 1]^A$. Therefore, $\forall \phi : \mathcal{A}_i \to \mathcal{A}_i$, $\xi_t^\phi := \left\langle \phi \circ \theta_i^{(t)} - \theta_i^{(t)}, u_i\left(\cdot, \theta_{-i}^{(t)}\right) - u_i^{(t)} \right\rangle$ is a bounded martingale difference sequence. By Azuma-Hoeffding inequality, for a fixed $\phi : \mathcal{A}_i \to \mathcal{A}_i$, with probability $1 - \delta'$,

$$\left| \sum_{t=1}^T \xi_t^\phi \right| \le 2\sqrt{2T \ln\left(\frac{2}{\delta'}\right)}.$$

By setting $\delta' = \delta/(NA^A)$, we get with probability $1 - \delta/N$, $\forall \phi : \mathcal{A}_i \to \mathcal{A}_i$,

$$\left| \sum_{t=1}^T \xi_t^\phi \right| \le 2\sqrt{2AT \ln\left(\frac{2AN}{\delta}\right)}.$$

Therefore we complete the proof by a union bound over $i \in [N]$. $\square$

*Proof of Theorem 12.* We condition on event $\Omega_3$ defined Lemma D.1, event $\Omega_4$ defined in Lemma D.4, and the success of Algorithm 1.

**Correlated Equilibrium.** By Lemma D.3 and Lemma D.4 we know that for all $i \in [N]$,

$$\text{SwapRegret}_T^{i,\star} \le O\left( \sqrt{A \ln(A)T} + \frac{A \ln(NAT/\Delta\epsilon)^2}{\Delta} + \sqrt{AT \ln\left(\frac{AN}{\delta}\right)} \right).$$

Therefore choosing

$$T = \Theta\left( \frac{A \ln\left(\frac{AN}{\delta}\right)}{\epsilon^2} + \frac{A \ln^3\left(\frac{NA}{\Delta\epsilon\delta}\right)}{\Delta\epsilon} \right)$$

will guarantee that $\text{SwapRegret}_T^{i,\star}$ is at most $\epsilon T/2$ for all $i \in [N]$. In this case the average strategy $(\sum_{t=1}^T \otimes_{i=1}^N \theta_i^{(t)})/T$ would be an $\epsilon/2$-CE.

Finally, in the clipping step, $\|\bar{\theta}_i^{(t)} - \theta_i^{(t)}\|_1 \le 2pA \le \frac{\epsilon}{4N}$ for all $i \in [N], t \in [T]$. Thus for all $t \in [T]$, we have $\| \otimes_{i=1}^n \bar{\theta}_i^{(t)} - \otimes_{i=1}^n \theta_i^{(t)} \|_1 \le \frac{\epsilon}{4}$, which further implies

$$\left\| (\sum_{t=1}^T \otimes_{i=1}^n \bar{\theta}_i^{(t)})/T - (\sum_{t=1}^T \otimes_{i=1}^n \theta_i^{(t)})/T \right\|_1 \le \frac{\epsilon}{4}.$$

Therefore the output strategy $\Pi = (\sum_{t=1}^{T} \otimes_{i=1}^{N} \bar{\theta}_i^{(t)})/T$ is an $\epsilon$-CE.

**Rationalizability.** By Lemma D.2, if $a \in E_L \cap \mathcal{A}_i$, $\theta_i^{(t)}(a) \leq p$ for all $t \in [T]$. It follows that $\bar{\theta}_i^{(t)}(a) = 0$, *i.e.*, the action would not be the support in the output strategy $\Pi = (\sum_t \otimes_i \bar{\theta}_i^{(t)})/T$.

**Sample complexity**. The total number of queries is

$$\sum_{i \in [N]} \sum_{t=1}^{T} AM_i^{(t)} \leq NAT + \sum_{i \in [N]} \sum_{b \in \mathcal{A}_i} \sum_{t=1}^{T} \frac{16\theta_i^{(t)}(b)}{\Delta^2 \cdot \sum_{\tau=1}^{t} \theta_i^{(\tau)}(b)}$$

$$\leq NAT + \frac{16NA^2}{\Delta^2} \cdot \ln(T/p)$$

$$\leq \widetilde{O}\left(\frac{NA^2}{\epsilon^2} + \frac{NA^2}{\Delta^2}\right),$$

where we used the fact that

$$\sum_{t=1}^{T} \frac{\theta_i^{(t)}(a)}{\sum_{i=1}^{\tau} \theta_i^{(\tau)}(a)} \leq 1 + \ln\left(\frac{T}{p}\right).$$

Finally consider the cost of finding one IDE-surviving action profile ($\widetilde{O}\left(\frac{LNA}{\Delta^2}\right)$) and we get the claimed rate. $\qquad\square$

## E   DETAILS FOR REDUCTION ALGORITHMS

In this section, we present the details for the reduction based algorithm for finding rationalizable CE (Algorithm 5) and analysis of both Algorithm 4 and 5.

### E.1   RATIONALIZABLE CCE VIA REDUCTION

We will choose $\epsilon' = \frac{\min\{\epsilon, \Delta\}}{3}$, $M = \left\lceil \frac{4\ln(2NA/\delta)}{\epsilon'^2} \right\rceil$.

**Lemma E.1.** *With probability $1 - \delta$, throughout the execution of Algorithm 4, for every $t$ and $i \in [N]$, $a_i' \in \mathcal{A}_i$,*

$$|\hat{u}_i(a_i', \Pi_{-i}) - u_i(a_i', \Pi_{-i})| \leq \epsilon'.$$

*Proof.* First, observe that during every iterate of $t$ before the algorithm returns, the total support size $\sum_{i=1}^{t} |\mathcal{A}_i^{(t)}|$ is increased by at least 1. It follows that the algorithm returns before $t = NA$.

By Hoeffding's inequality,

$$\Pr\left[|\hat{u}_i(a_i', \Pi_{-i}) - u_i(a_i', \Pi_{-i})| > \epsilon'\right] \leq 2\exp\left(-\frac{n\epsilon'^2}{2}\right) \leq \frac{\delta}{N^2 A^2}.$$

Applying union bound over $t$, $i$ and $a_i'$ proves the statement. $\qquad\square$

*Proof of Theorem 13.* **Correctness.** Since $\Pi$ is an $\epsilon$-CCE in the subgame $\Pi_{i=1}^{N} \mathcal{A}_i^{(t)}$, $\forall i \in [N]$, $\forall a \in \mathcal{A}_i^{(t)}$

$$u_i(a, \Pi_{-i}) \leq u_i(\Pi) + \epsilon'.$$

Because $\arg\max_{a \in \mathcal{A}_i} \hat{u}_i(a, \Pi_{-i}) \in \mathcal{A}_i^{(t)}$, $\forall i \in [N]$, $\forall a \in \mathcal{A}_i$

$$u_i(a, \Pi_{-i}) \leq \hat{u}_i(a, \Pi_{-i}) + \epsilon' \leq \max_{a' \in \mathcal{A}_i^{(t)}} \hat{u}_i(a', \Pi_{-i}) + \epsilon'$$

$$\leq \max_{a' \in \mathcal{A}_i^{(t)}} u_i(a', \Pi_{-i}) + 2\epsilon'$$

$$\leq u_i(\Pi) + 3\epsilon' \leq u_i(\Pi) + \epsilon.$$

Therefore $\Pi$ is an $\epsilon$-CCE in the full game.

Moreover, we claim that for any $t$, $\mathcal{A}_i^{(t)}$ only contains $\Delta$-rationalizable actions. This is true for $t = 1$ with high probability due to our initialization. Suppose that this is true for $t$. Notice that the only way for an action $a_i' \in \mathcal{A}_i^{(t+1)}$ is to be an empirical best response, which means

$$u_i(a_i', \Pi_{-i}) \geq \hat{u}_i(a_i', \Pi_{-i}) - \epsilon' \geq \max_{a \in \mathcal{A}_i} \hat{u}_i(a, \Pi_{-i}) - \epsilon'$$

$$\geq \max_{a \in \mathcal{A}_i} u_i(a, \Pi_{-i}) - 2\epsilon'.$$

Since $\epsilon' < \Delta/2$, this means that $a_i'$ is the $\Delta$-best response to a $\Delta$-rationalizable strategy, and is therefore $\Delta$-rationalizable. Therefore $\mathcal{A}_i^{(t+1)}$ also only contains $\Delta$-rationalizable actions. Our claim can be thus proven via induction, and it follows that the output strategy is also $\Delta$-rationalizable.

We conclude that the output strategy is a $\Delta$-rationalizable $\epsilon$-CCE with probability $1 - 2\delta$ (assuming the event in Lemma E.1 as well as the rationalizability of the initialization).

**Sample complexity.** By Theorem 3, Line 1 needs $\widetilde{O}\left(\frac{LNA}{\Delta^2}\right)$ samples. Since the algorithm returns before $t = NA$, the total number of calls to the black-box oracle $\mathcal{O}$ is $NA$. For each $t$, the number of samples required is

$$NAM = \widetilde{O}\left(\frac{NA}{\min\{\Delta, \epsilon\}^2}\right).$$

Combining this with the upper bound on $t$, and the cost for Algorithm 1 gives the total sample complexity bound

$$\widetilde{O}\left(\frac{N^2 A^2}{\min\{\Delta^2, \epsilon^2\}}\right).$$

$\square$

## E.2 RATIONALIZABLE CE VIA REDUCTION

The algorithm for CE is quite similar to the one for CCE, except now when testing whether a subgame $\epsilon$-CE is an actual $\epsilon$-CE, we need to use the conditional distribution $\Pi|a_i$, which is the conditional distribution of the other players' actions given that player $i$ is told to play $a_i$. The detailed description is given in Algorithm 5. Similar to the CCE case, we will choose $\epsilon' = \frac{\min\{\epsilon, \Delta\}}{3}$, $M = \left\lceil \frac{4\ln\left(2NA^2/\delta\right)}{\epsilon'^2} \right\rceil$.

---

**Algorithm 5** Rationalizable $\epsilon$-CE via Black-box Reduction

1: $(a_1^\star, \cdots, a_N^\star) \leftarrow$ `Algorithm 1`
2: For all $i \in [N]$, initialize $\mathcal{A}_i^{(1)} \leftarrow \{a_i^\star\}$ for all $i \in [N]$
3: **for** $t = 1, 2, \ldots$ **do**
4:     Find an $\epsilon'$-CE, $\Pi$, in the sub-game supported on $\Pi_{i \in [N]}\mathcal{A}_i^{(t)}$
5:     $\forall i \in [N], a_i, a_i' \in \mathcal{A}_i$, sample $u_i(a_i', \Pi_{-i}|a_i)$ for $M$ times and compute average $\hat{u}_i(a_i', \Pi_{-i}|a_i)$

6:     **for** $i \in [N]$ **do**
7:         **for** $a_i \in \mathcal{A}_i^{(t)}$ **do**
8:             Let

$$a_i' \leftarrow \arg\max_{a \in \mathcal{A}_i} \hat{u}_i(a, \Pi|a_i) \text{// Computing the empirical best response}$$

9:         $\mathcal{A}_i^{(t+1)} \leftarrow \mathcal{A}_i^{(t)} \cup \{a_i'\}$
10:    **if** $\mathcal{A}_i^{(t)} = \mathcal{A}_i^{(t+1)}$ for all $i \in [N]$ **then**
11:       **return** $\Pi$

---

**Lemma E.2.** *With probability $1 - \delta$, throughout the execution of Algorithm 4, for every $t$ and $i \in [N]$, $a_i' \in \mathcal{A}_i$, $a_i \in \mathcal{A}_i$,*

$$|\hat{u}_i(a_i', \Pi|a_i) - u_i(a_i', \Pi|a_i)| \leq \epsilon'.$$

*Proof.* First, observe that during every iterate of $t$ before the algorithm returns, the total support size $\sum_{i=1}^{t} |\mathcal{A}_i^{(t)}|$ is increased by at least 1. It follows that the algorithm returns before $t = NA$.

By Hoeffding's inequality,

$$\Pr\left[|\hat{u}_i(a_i', \Pi|a_i) - u_i(a_i', \Pi|a_i)| > \epsilon'\right] \leq 2\exp\left(-\frac{n\epsilon'^2}{2}\right) \leq \frac{\delta}{N^2 A^3}.$$

Applying union bound over $t$, $i$, $a_i$, and $a_i'$ proves the statement. $\qquad\square$

*Proof.* Note that with high probability, the empirical estimates $\hat{U}$ are at most $\epsilon/4$ away from the true value $U$. Since $a_i'$ is the empirical best response, we have

$$U_i(a_i', \Pi|a_i) \geq \arg\max_{a \in \mathcal{A}_i} U_i(a, \Pi|a_i) - \epsilon.$$

Note that $\Pi|a_i$ is supported on actions that can survive any rounds of $\epsilon$-IDE. Therefore it serves as a certificate that $a_i'$ will never be $\epsilon$-eliminated as well. $\qquad\square$

**Lemma E.3.** *The returned strategy $\Pi$ is an $\epsilon$-CE with probability $1 - \delta$.*

*Proof.* When the algorithm terminates, for all $i \in [N]$,

$$\sum_{a_i \in \mathcal{A}_i^{(t)}} \Pi_i(a_i) \cdot \left(\max_{a \in \mathcal{A}_i} \hat{u}_i(a, \Pi|a_i) - \max_{a \in \mathcal{A}_i^{(t)}} \hat{u}_i(a, \Pi|a_i)\right) = 0.$$

Therefore

$$\sum_{a_i \in \mathcal{A}_i^{(t)}} \Pi_i(a_i) \cdot \left(\max_{a \in \mathcal{A}_i} u_i(a, \Pi|a_i) - \max_{a \in \mathcal{A}_i^{(t)}} u_i(a, \Pi|a_i)\right) \leq 2\epsilon'.$$

Since $\Pi$ is an $\epsilon'$-CE in the reduced game,

$$\sum_{a_i \in \mathcal{A}_i^{(t)}} \Pi_i(a_i) \cdot \left(\max_{a \in \mathcal{A}_i^{(t)}} u_i(a, \Pi|a_i) - u_i(a_i, \Pi|a_i)\right) \leq \epsilon'.$$

Summing the two inequalities above gives

$$\sum_{a_i \in \mathcal{A}_i^{(t)}} \Pi_i(a_i) \cdot \left(\max_{a \in \mathcal{A}_i} u_i(a, \Pi|a_i) - u_i(a_i, \Pi|a_i)\right) \leq 3\epsilon',$$

which proves the statement. $\qquad\square$

**Lemma E.4.** *For any $t$, $\mathcal{A}_i^{(t)}$ only contains $\Delta$-rationalizable actions with probability $1 - 2\delta$.*

*Proof.* We prove this inductively. This is true for $t = 1$ with probability $1 - \delta$ due to our initialization. Suppose that this is true for $t$. Notice that the only way for an action $a_i' \in \mathcal{A}_i^{(t+1)}$ is to be an empirical best response, which means for some $a_i$

$$u_i(a_i', \Pi|a_i) \geq \hat{u}_i(a_i', \Pi|a_i) - \epsilon' \geq \max_{a \in \mathcal{A}_i} \hat{u}_i(a, \Pi|a_i) - \epsilon'$$

$$\geq \max_{a \in \mathcal{A}_i} u_i(a, \Pi|a_i) - 2\epsilon'.$$

Since $\epsilon' < \Delta/2$, this means that $a_i'$ is the $\Delta$-best response to a $\Delta$-rationalizable strategy, and is therefore $\Delta$-rationalizable. Therefore $\mathcal{A}_i^{(t+1)}$ also only contains $\Delta$-rationalizable actions. Our claim can be thus proven via induction, and it follows that the output strategy is also $\Delta$-rationalizable. $\qquad\square$

*Proof of Theorem 14.* **Correctness.** By Lemma E.3 and E.4, the output strategy is a $\Delta$-rationalizable $\epsilon$-CE with probability $1 - 2\delta$ (assuming that the event in Lemma E.2 holds and the rationalizability of the initialization).

**Sample complexity.** The total sample complexity is

$$\widetilde{O}\left(\frac{LNA}{\Delta^2}\right) + NA \times NA^2M = \widetilde{O}\left(\frac{N^2A^3}{\min\{\Delta, \epsilon\}^2}\right).$$

$\square$

