# OpenReview forum: "Learning Rationalizable Equilibria in Multiplayer Games"
_ICLR.cc/2023/Conference — ICLR 2023 poster_

### Official Review · Reviewer_xdmb · 2022-10-24

**Confidence:** 2
**Correctness:** 3
**Technical Novelty And Significance:** 3
**Empirical Novelty And Significance:** 3
**Recommendation:** 8

**Clarity, Quality, Novelty And Reproducibility:**

Clarity: the paper is very well written and easily understandable.
Quality: the results seem correct.
Novelty: sufficient.
Reproducibility: this work does not contain any experimental results.


**Strength And Weaknesses:**

I find this work very well written in terms of both quality of prose and the overall exposition. All notions the authors introduce are well described, and the notation is clear and easily comprehensible. The paper also competently explains how the presented work fits into the literature on the topic. The theory seems solid, I briefly went through the proofs in the appendix and as far as I can tell, they apper to be correct. I especially liked the black-box techniques the authors present.

What I am missing a bit in the main text is some more in-depth analysis of the motivation that would highlight the importance of this work, perhaps some illustrative examples … something along the lines of showing what could actually happen in practice if not rationalizable (C)CEa are adopted? What is a possible degradation in terms of utility? I assume I may find some analysis in the cited works, but I can’t shake the feeling of being slightly uncertain of how important this problem truly is.

My other concern relates to the potential uses of the derived sample complexities and how to estimate them in practice. If I understand this work correctly, the authors work in the bandit setting when the underlying game is unknown. While N and A are given, and delta could be chosen, how could the “L” be computed in the bandit case, given that you need it to run Algorithm 1 (line 2)? Even if the original utilities are fully observable, how difficult is it to compute “L”? If I am not wrong, this definition of strict dominance is with respect to mixed strategies, so calculating “L” corresponds to iteratively resolve the linear programs from the original work by Vincent Conitzer. How difficult is it in contrast to the algorithms presented in this paper? Or is “L” not supposed to be computed in practice? If yes, are there some reasonable bounds on “L” that would allow us to estimate the expected solution quality given some supply of samples?

Finally, just a few nits:
(i) The introduction presents the sample complexities without explaining at all what delta is.
(ii) The delta symbol is then further used both as the “reward gap” in the definition of rationalizability and to denote distributions over some set … Is there some connection between the two that would justify the use of the same notation that I am missing?
(iii) FTRL remains undefined on page 2, I assume this is the follow-the-regularized-leader.


**Summary Of The Paper:**

This paper studies learnability of (coarse) correlated and Nash equilibria supported on rationalizable actions in unknown multiplayer normal-form games. After introducing the relevant concepts, the authors derive the first main result: a probabilistic bandit algorithm that identifies a rationalizable action profile using a number of samples inversely quadratic in an elimination reward gap and linear in the number of players (N), maximum number of actions (A), and a minimum number of iterations to eliminate dominated actions. This result serves as a stepping stone to the next two algorithms, finding rationalizable correlated and coarse correlated equilibria with sample complexities extending the action-profile-algorithm with additional (at most quadratic) terms. As a side-product, the authors obtain also a sample complexity of finding rationalizable Nash equilibria in two-player zero-sum games. The last main result consists of two algorithms capable of “rationalizing” an arbitrary black-box algorithm for computing correlated or coarse correlated equilibrium using N*A calls to the black-box and a number of additional samples at most quadratic (in case of coarse correlation) or cubic (in case of standard correlation) in the game’s parameters.

**Summary Of The Review:**

A technically solid paper that may benefit from a more detailed discussion on the consequences of (ir)rationalizability, but otherwise deserves publication, in my opinion.

---

> ### Author Response · Authors · 2022-11-12
> **Response to Reviewer xdmb**
>
> We thank the reviewer for their valuable feedback. We respond to the main concerns below.
>
> —"*What I am missing a bit in the main text is some more in-depth analysis of the motivation that would highlight the importance of this work, perhaps some illustrative examples … something along the lines of showing what could actually happen in practice if not rationalizable (C)CEa are adopted?*"
>
> A general-sum game admits multiple CEs/CCEs, and rationalizability could be a sensible property to additionally satisfy. In fact, rationalizability models human behavior in games: experiments in games such as the famous “beauty contest” (due to Keynes, 1936) suggest that actual human participants will converge toward rationalizable actions in repetitive gameplay (see Ch. 1.2.3, Camerer 2011).
>
> For a concrete example, consider the following 4x4 two-player game and its CCE (example inspired by Viossat and Zapechelnyuk, 2013):
> |Actions\Actions |$b_1$|$b_2$|$b_3$|$b_4$|
> |----|---|---|---|---|
> |$a_1$|$1,1$|$1,1$|$1,0$|$5,0$|
> |$a_2$|$1,1$|$1,1$|$5,0$|$1,0$|
> |$a_3$|$0,1$|$0,5$|$4,4$|$0,0$|
> |$a_4$|$0,5$|$0,1$|$0,0$|$4,4$|
>
> , and the CCE{0.5: $(a_3,b_3)$, 0.5: $(a_4,b_4)$}. It can be seen that this CCE is entirely supported on *dominated actions*, and it would be hard to, for instance, persuade players to carry out this correlated policy. Rationalizable CCEs will naturally rule out this particular CCE.
>
>
> —"*While N and A are given, and delta could be chosen, how could the “L” be computed in the bandit case, given that you need it to run Algorithm 1 (line 2)? [...] Or is “L” not supposed to be computed in practice? If yes, are there some reasonable bounds on “L” that would allow us to estimate the expected solution quality given some supply of samples?*"
>
> We thank the reviewer for bringing up this issue. When the exact value of $L$ is unknown, we can use its maximal value $NA$ in the algorithm, which only results in at most a multiplicative factor of $NA$ in the sample complexity. However, determining the exact value of $L$ (the minimum elimination length) is statistically hard in the bandit feedback setting. For example, from the proof of Proposition 2, even deciding whether $L=0$ or $1$ can take exponentially many samples in the worst case.
>
> Nevertheless, as the reviewer pointed out, we can instantiate our algorithm with any guessed $L’$ in practice. If $L’$ is too conservative (e.g., chosen to be $NA$), the sample complexity would scale with $L’$ and could be potentially loose. If $L’$ is chosen too small, we can still guarantee that equilibria produced by the algorithm avoid actions in $E_{L’}$, a weaker yet non-trivial guarantee.
>
>
> —"*Finally, just a few nits: (i) The introduction presents the sample complexities without explaining at all what delta is. (ii) The delta symbol is then further used both as the “reward gap” in the definition of rationalizability and to denote distributions over some set … Is there some connection between the two that would justify the use of the same notation that I am missing? (iii) FTRL remains undefined on page 2, I assume this is the follow-the-regularized-leader.*"
>
> We thank the reviewer for pointing them out. We slightly abuse the notation to use $\Delta$ to refer to both gap and simplex, which are both the usual notational tradition in the bandits/online learning literature. We have added a footnote in our revision to avoid confusions. We have also incorporated the other suggestions about clarity in our revision.
>
> [Reference]
>
> Colin F Camerer. Behavioral game theory: Experiments in strategic interaction. Princeton university press, 2011
>
> J Keynes. The General Theory of Employment, Interest and Money. 1936.

---

### Official Review · Reviewer_eq1u · 2022-10-25

**Confidence:** 3
**Correctness:** 4
**Technical Novelty And Significance:** 3
**Empirical Novelty And Significance:** 2
**Recommendation:** 8

**Clarity, Quality, Novelty And Reproducibility:**

**Clarity:**
- I think the paper is well-written overall. The introduction of the related concepts is reader-friendly and the arguments are easy to follow.

**Strength And Weaknesses:**

**Strength:**
- The authors have proposed a set of algorithms that solves the problem of learning rationalizable equilibrium, which is an important and interesting task.
- The sample complexity of the proposed method for finding rationalizable action profiles improves significantly compared to the existing Exp3-DH algorithm, and a corresponding lower bound is provided, which shows the near optimality of the algorithm under $L=O(1)$.
- The paper offers a framework to handle the task of finding rationalizable equilibria by augmenting arbitrary existing algorithms that find the equilibria. Even though there is a slight sacrifice in sample complexity, such a framework can be useful when combined with equilibria-finding algorithms of different properties.

**Weakness:**
- The matching lower bounds for the learning algorithms on CE and CCE are still missing.

**Summary Of The Paper:**

This paper provides an algorithm for learning the rationalizable action profiles for multi-agent games with bandit feedback, based on which another algorithm for learning the approximate rationalizable CE and CCE. Both algorithms are computationally efficient and have been proven to improve significantly compared to prior work in terms of sample complexity.

**Summary Of The Review:**

I feel like this is a good paper with novel ideas and solid proof. It considers a critical problem of finding rationalizable equilibria and brings sample-efficient algorithms with theoretical guarantees for multi-agent general-sum games under bandit feedback.

================================

The authors addressed all my questions, and I tend to keep my original score.

---

> ### Author Response · Authors · 2022-11-12
> **Response to Reviewer eq1u**
>
> We thank the reviewer for their valuable feedback. We respond to the main concerns below.
>
> —"*The matching lower bounds for the learning algorithms on CE and CCE are still missing.*"
>
> We can naively obtain the following lower bound for learning $\Delta$-rationalizable $\epsilon$-CCE/CE by combining Theorem 4 with existing lower bounds for CCE (see, e.g., Mannor and Tsitsiklis 2004): $\Omega\left(\frac{NA}{\Delta^2}+\frac{A}{\epsilon^2}\right)$.
>
> Currently, our upper bound (Theorem 6) does not match the above lower bound even when $L=O(1)$. As mentioned in Remark 7, closing this gap might require more sophisticated exploration methods and/or utility estimators, which we leave as future work.
>
> [Reference]
>
> Mannor and Tsitsiklis. The Sample Complexity of Exploration in the Multi-Armed Bandit Problem. JMLR, 2004.

---

### Official Review · Reviewer_ta5R · 2022-10-25

**Confidence:** 4
**Correctness:** 4
**Technical Novelty And Significance:** 3
**Empirical Novelty And Significance:** Not applicable
**Recommendation:** 8

**Clarity, Quality, Novelty And Reproducibility:**

CLARITY: The paper is well written and fairly easy to follow.

QUALITY: As far as I am concerned, the technical results are sound.

NOVELTY: The problem faced by the paper has not been addresses in the literature yet. The technical novelty of the results is not clear, the authors should do a better job in "selling" their technical contributions.

REPRODUCIBILITY: I believe the proofs can be easily reproduced. There are no experiments.

**Strength And Weaknesses:**

STRENGTHS

1) The paper studies an interesting problem that is largely overlooked in the literature. While most of the papers focus on learning equilibria in games, with the goal of finding AN (approximate) equilibrium, I believe that focusing on how to learn equilibria with desirable properties is a crucial property for operationalizing equilibrium-learning algorithms in practice.

2) The paper is well written and fairly easy to follow. As far as I am concerned, the technical results are sound.

WEAKNESSES

1) The paper addresses a very specific topic that could be interesting only for a very narrow parto of the ICLR community.

2) The technical tools used to prove the results seem quite standard, or perhaps the authors did not stress sufficiently the technical novelty of their results.

MINOR ISSUES

1) The conditioning inside the expectation at the beginning of Section 2 is not clear.
2) The notation u_i(x_i,x_{-i)) and u_i(x_i,x_{-I}) has not been formally introduced.
3) It is not clear how to set L in the algorithms, do you always have to set it to be equal to its maximum value N x A?
4) The notation with \theta inside Algorithms 2 and 3 is not clear, I had to re-read them several times in order to grasp the meaning. I think it is better to introduce it in the main text, together with a short description in words.
5) I would add formal definitions of \Delta-rationalizable \epsilon-CE and \epsilon-CCE.

**Summary Of The Paper:**

This paper studies the problem of designing no-regret learning algorithms that provably converge to Coarse Correlated Equilibria (CCEs) and Correlated Equilibria (CEs) that avoid playing actions that are non-rationalizable, i.e., they do not survive iterative elimination od dominated actions. The paper focuses on normal-form games with stochastic payoffs, under the bandit feedback model. First, it provides an algorithm that can be used to learn any rationalizable action profile. Then, it uses such an algorithm as a subroutine to design sample-efficient no-regret learning algorithms for CCEs and CEs. The paper provides two variants of such algorithms: one is based on a modification of the Hedge algorithm, while the other works by using any standard no-regret algorithms as a subroutine, at the expense of achieving "slightly" worse regret bounds. The paper also provides a lower bound showing that the sample complexity of the first variant of the algorithms is optimal with respect to the number of players and the number of actions.

**Summary Of The Review:**

I think that this is a good paper, even tough the topic addressed by it is rather narrow. My only concerns are about the novelty of the technical results.

---

> ### Author Response · Authors · 2022-11-12
> **Response to Reviewer ta5R**
>
> We thank the reviewer for their valuable feedback. We respond to the main concerns below.
>
> —"*The technical tools used to prove the results seem quite standard, or perhaps the authors did not stress sufficiently the technical novelty of their results.*"
>
> While our main algorithm is based on the classical algorithm of Hedge, it incorporates several novel components, such as initializing with a rationalizable action profile, and variance reduction via mini-batch sampling. These techniques combine to enhance Hedge with interesting properties such as guarantees near-rationalizability in every iterate. We believe such proof technique is new, and has not appeared in the standard analysis. Please also see our discussion in Section 4.1.1.
>
>
> —"*It is not clear how to set L in the algorithms, do you always have to set it to be equal to its maximum value N x A?*"
>
> We agree that when the exact value of $L$ is unknown, one way to set it is to default to its maximal value $NA$. Nevertheless, we can set $L$ to be any guessed value $L’$ in practice. If $L’$ is too conservative (e.g., chosen to be $NA$), the sample complexity would scale with $L’$  and could be potentially loose. If $L’$ is chosen too small, we can still guarantee that equilibria produced by the algorithm avoid actions in $E_{L’}$, a weaker yet non-trivial guarantee.
>
>
> —"*Other minor issues about clarity*"
>
> We thank the reviewer for pointing out these issues. We have addressed them in the revision.

---

### Official Review · Reviewer_TWWs · 2022-11-01

**Confidence:** 4
**Correctness:** 4
**Technical Novelty And Significance:** 2
**Empirical Novelty And Significance:** Not applicable
**Recommendation:** 6

**Clarity, Quality, Novelty And Reproducibility:**

I believe the paper is relatively well written.
The question that is examined here was introduced recent in the work of Wu et al. This work is effectively a direct follow-up.



**Strength And Weaknesses:**

Strengths
1) The paper makes recent progress on a paper about learning in games
2) The writing is relatively clear

Weaknesses
1) The motivation behind finding rationalizable CCE does not seem to be particularly strong. Why is so important for the players to converge to a rationalizable CE/CCE? The proposed algorithms are rather unnatural and although can be used to compute CCE do not provide no-regret guarantees themselves. Hence, one would need a very strong justification of why this specific class of equilibria are particularly desirable. Also, if "natural" learning algorithms like Hedge and variants do not reproduce iteratively rationalizable CCE fast isn't this an indication that maybe iterative rationalizability is not a "natural" property?
2) The algorithms are rather impractical to use and result in significant blow-ups e.g. the sampling complexity grows linearly in the number of agents whereas without it there is no dependence on the number of agents.
3) There are no proposed interesting practical applications of this idea (also no experimental section)

**Summary Of The Paper:**

The paper presents algorithms for computing approximately rationalizable correlated equilibria (CE) and coarse correlated equilibria (CCE) in general games. Although these bounds are not shown to be optimal (except of the case where strategies are rationalizable after a constant number of iterated elimination rounds) they are significantly improved over the recent work of Wu et al. Finally the paper provides reduction schemes that find approximately-rationalizable epsilon-CCE/CE using black-box algorithms for epsilon-CE/CCE.

**Summary Of The Review:**

A theoretical paper that makes progress on the subject of rationalizability in equilibria. The motivation/applications are somewhat lacking.

---

> ### Author Response · Authors · 2022-11-12
> **Response to Reviewer TWWs**
>
> We thank the reviewer for their valuable feedback. We respond to the main concerns below.
>
> We would like to first clarify that we are not a direct follow-up to Wu et al.: the EXP3-DH algorithm in Wu et al. is only guaranteed to find strategies that are rationalizable but not necessarily equilibria. Our paper provides the first polynomial sample efficiency guarantees for finding rationalizable equilibria.
>
> —"*The motivation behind finding rationalizable CCE does not seem to be particularly strong. Why is so important for the players to converge to a rationalizable CE/CCE? […] Hence, one would need a very strong justification of why this specific class of equilibria are particularly desirable.*"
>
> - Rationalizability is a well-accepted solution concept in the field of game theory (see e.g. Ch. 4 Osborne and Rubinstein 1994, Bernheim 1984). It is a criterion that Nash Equilibrium and exact Correlated Equilibrium satisfy, but their relaxations (approximate CE and CCE) do not.
> - Rationalizability explains human behavior in games: experiments in games such as the famous “beauty contest” (due to Keynes, 1936) suggest that actual human participants will converge toward rationalizable actions in repeated gameplay (see Ch. 1.2.3, Camerer 2011).
> - A general game admits multiple CEs/CCEs, with some of them good and some of them bad. Rationalizability could be a sensible property to additionally satisfy. For example, consider the following 4x4 two-player game (example inspired by Viossat and Zapechelnyuk, 2013):
> |Actions\Actions |$b_1$|$b_2$|$b_3$|$b_4$|
> |----|---|---|---|---|
> |$a_1$|$1,1$|$1,1$|$1,0$|$5,0$|
> |$a_2$|$1,1$|$1,1$|$5,0$|$1,0$|
> |$a_3$|$0,1$|$0,5$|$4,4$|$0,0$|
> |$a_4$|$0,5$|$0,1$|$0,0$|$4,4$|
>
> , and the CCE{0.5: $(a_3,b_3)$, 0.5: $(a_4,b_4)$}. It can be seen that this CCE is entirely supported on *dominated actions*, and it would be hard to, for instance, persuade players to carry out this correlated policy. Rationalizable CCEs will naturally rule out this particular CCE.
>
>
> —"*The proposed algorithms are rather unnatural and although can be used to compute CCE do not provide no-regret guarantees themselves. [...] Also, if "natural" learning algorithms like Hedge and variants do not reproduce iteratively rationalizable CCE fast isn't this an indication that maybe iterative rationalizability is not a "natural" property?*"
>
> For the rationalizability property, we believe its “naturalness” could be rather subjective; and whether Hedge (or its variants) achieves it may not be the only standard for its “naturalness”. Besides, we believe our algorithm (e.g., Algorithm 2) can be indeed viewed as a variant of Hedge, with a special initialization and a natural loss estimator that handles bandit feedback.
>
> We admit that the current algorithm does not have a no-regret guarantee, which is an interesting question for future investigation.
>
> —"*The algorithms are rather impractical to use and result in significant blow-ups e.g. the sampling complexity grows linearly in the number of agents whereas without it there is no dependence on the number of agents.*"
>
> Indeed, different from learning CCEs, a sample complexity at least linear in the number of agents is necessary to establish any rationalizability guarantee, as indicated by our lower bound (Theorem 4).
>
> — "*Lack of interesting practical applications of this idea (also no experimental section).*"
>
> Our solution concept and proposed algorithm are general-purpose in the sense that they can be used in any application that can be modeled as a normal-form game. We note that the focus of this paper is mainly in theory, while we agree that experiments are important directions for future work.
>
> [Reference]
>
> Martin J Osborne and Ariel Rubinstein. A course in game theory. MIT press, 1994.
>
> B Douglas Bernheim. Rationalizable strategic behavior. Econometrica. 1984.
>
> Colin F Camerer. Behavioral game theory: Experiments in strategic interaction. Princeton university press, 2011
>
> J Keynes. The General Theory of Employment, Interest and Money. 1936.

---

> > ### Comment · Reviewer_TWWs · 2022-11-24
> > **Response**
> >
> > I thank the authors for their detailed response. This address some of my issue hence I slightly increase my score accordingly.

---

### Decision · Program_Chairs · 2023-01-20

**Decision:**

Accept: poster

**Justification For Why Not Higher Score:**

The concerns about whether this is a well-motivated problem were not well addressed in my view.  Learning equilibria with desirable properties is an important direction, but it is less clear whether "rationalizable" is a particularly desirable property (for instance, there are games in which the highest joint payoffs are achieved when all agents play unrationalizable strategies).


**Justification For Why Not Lower Score:**

The reviewers were all rather positive about the paper in spite of its motivation issues; I think it deserves to be accepted in some form.

**Metareview: Summary, Strengths And Weaknesses:**

(a) Summary:  This paper proposes algorithms for efficiently computing rationalizable correlated and coarse-correlated equilibria in general-sum games.

(b) Strengths: This paper makes clear progress on the subject of learning equilibria with desirable properties.  All the reviewers found the writing clear and the contribution easy to understand.

(c) Weaknesses: Multiple reviewers had concerns about the motivation of this work; bluntly, is this an important problem to solve?  There were also questions about the novelty of the approach.


**Note From Pc:**

if the above contains the word "oral" or "spotlight" please see: "oral" presentation means -> notable-top-5% and "spotlight" means -> notable-top-25%. As stated in our emails, we are disassociating presentation type from AC recommendations

**Summary Of Ac-Reviewer Meeting:**

n/a